# ARBITRARY VIRTUAL TRY-ON NETWORK: CHARACTERISTICS REPRESENTATION AND TRADE-OFF BETWEEN BODY AND CLOTHING

**Yu Liu, Mingbo Zhao**[*]
Donghua University, China
{2191408,mzhao4}@dhu.edu.cn

**Zhao Zhang**[*]
Hefei University of Technology, China
cszzhang@gmail.com

**Jicong Fan**
The Chinese University of Hong Kong, Shenzhen, China
fanjicong@cuhk.edu.cn

**Yang Lou**
Osaka University, Japan
felix.lou@ieee.org

**Shuicheng Yan**
Sea AI Lab, Singapore
yansc@sea.com

## ABSTRACT

Deep learning based virtual try-on system has achieved some encouraging progress recently, but there still remain several big challenges that need to be solved, such as trying on arbitrary clothes of all types, trying on the clothes from one category to another and generating image-realistic results with few artifacts. To handle this issue, we propose the Arbitrary Virtual Try-On Network (AVTON) that is utilized for all-type clothes, which can synthesize realistic try-on images by preserving and trading off characteristics of the target clothes and the reference person. Our approach includes three modules: 1) Limbs Prediction Module, which is utilized for predicting the human body parts by preserving the characteristics of the reference person. This is especially good for handling cross-category try-on task (e.g., long sleeves ↔ short sleeves or long pants ↔ skirts, etc.), where the exposed arms or legs with the skin colors and details can be reasonably predicted; 2) Improved Geometric Matching Module, which is designed to warp clothes according to the geometry of the target person. We improve the TPS-based warping method with a compactly supported radial function (Wendland's $\Psi$-function); 3) Trade-Off Fusion Module, which is to trade off the characteristics of the warped clothes and the reference person. This module is to make the generated try-on images look more natural and realistic based on a fine-tuning symmetry of the network structure. Extensive simulations are conducted and our approach can achieve better performance compared with the state-of-the-art virtual try-on methods.

## 1 INTRODUCTION

During the past few years, computer vision technology has been widely utilized in the extensive applications of artificial fashion. These applications include clothes detection Liu et al. (2016); Ge et al. (2019), clothes parsing Li et al. (2019); Gong et al. (2017), clothes attributions and categories

---

[*]Corresponding authors. This work is supported by the National Natural Science Foundation of China (61971121, 61672365, 62106211). Our code and dataset are available at https://github.com/LiuYuZzz/AVTON.

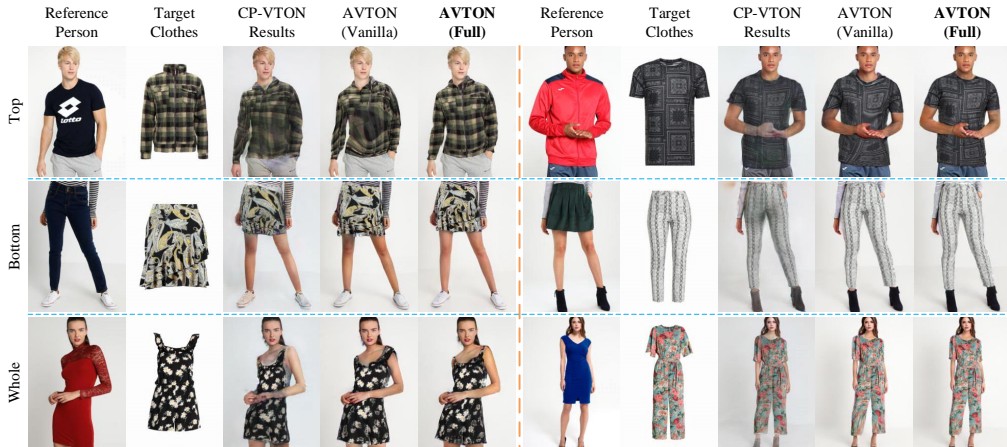

Figure 1: We propose AVTON that is trained with an *all-type clothing dataset*. It can be adapted to all-type clothing try-on tasks and get image-realistic results. The types of clothes are divided into the *top, bottom, and whole*, and we apply CP-VTON Wang et al. (2018a) and AVTON on them. The CP-VTON is retained with the all-type clothing dataset, and the AVTON (Vanilla) indicates the AVTON trained without *LPM and Wendland's Ψ-function* Wendland (1995), while the AVTON (Full) is just the opposite.

recognition Wang et al. (2018a); Liu et al. (2016); Ge et al. (2019); Gao et al. (2022); Zeng et al. (2022), clothes collocation Iwata et al. (2011); Zhao et al. (2020); He et al. (2016); Shih et al. (2018); Li et al. (2017); Han et al. (2017); Cui et al. (2019); Zeng et al. (2020), etc. These applications are all merited from recently developed technology, namely deep learning, due to its powerful feature extraction ability to capture rich mid-level image representations. Motivated by this technology, deep learning-based virtual try-on methods have been extensively studied and achieved considerable results recently.

Many deep learning-based methods for virtual try-on have been developed during the past decade. VITON Han et al. (2018) and CP-VTON Wang et al. (2018a) are the two first works. The key idea of VITON is to exploit a Thin-Plate Spline (TPS) Duchon (1977) based method to warp the clothes images with texture mapped on it, while CP-VTON has extended VITON by developing neural network layers to learn the transformation parameters. VTNFP Yu et al. (2019) and ACGPN Yang et al. (2020) are another two popular methods that have improved the performance of the try-on task with more characteristics of clothes and human body preserved. Recently, many methods have been developed that focus on a certain aspect to improve performance. For example, the PFAFN Ge et al. (2021) adopts the knowledge distillation from a large complex network to achieve the faster reference time as well as remove the human parsing procedure; VITON-HD method Choi et al. (2021); Lee et al. (2022) is specially developed to handle high-resolution try-on task. Dong et al. (2022) further proposes wFlow for handling video try-on tasks in the wild by adopting the optical flow from videos, while Jiang et al. (2022) has firstly utilized transformer for taming video try-on tasks. All the above methods have verified the effectiveness of deep learning-based virtual try-on.

Although these works have made some progress, most of them only focus on top cloth try-on tasks and cannot handle arbitrary try-on tasks (the top, bottom, or the whole clothes). This is of great practice in real-world applications and currently, there are few works focused on this try-on task. In addition, it still remains some ongoing challenges and limitations: 1) most benchmark datasets utilized for training virtual try-on methods mainly contain top clothes. As a result, the trained model can only handle the top clothing try-on task while cannot be adapted to work on the other bottom or whole clothing try-on task; 2) cross-category try-on task (e.g., long sleeves↔short sleeves or long pants↔skirts, etc.) is another challenge in the try-on task. A case in point is that when people aim to try on from long sleeves to short sleeves, some parts of people's arms will be exposed. Therefore, it is necessary to preserve the characteristics of the reference person and predict such an exposed human body when generating the image-realistic try-on results. However, most current methods Han et al. (2018); Wang et al. (2018a); Pandey & Savakis (2020); Yu et al. (2019) do not consider this issue. As a result, some bad try-on performances, e.g., the limbs of human beings are covered by clothes,

the color of the skin is wrongly painted and the hand details cannot be properly generated, may be appeared; 3) current methods are not good at trading off characteristics of the warped clothes and the reference person. Although it can preserve the characteristics of the warped cloth and reference person as much as possible, its generated images are not realistic enough, such as some artifacts near the neck regions. The reason for these results is that the final fusion module prefers to preserve the warped clothing characteristics and cannot correct these errors.

In this paper, we propose a new image-based virtual try-on method, called AVTON, to achieve arbitrary clothing try-on and image-realistic results. The proposed method contains three modules: a) Limbs Prediction Module, which is developed for predicting limbs, and keep the head and the non-target human body parts, to preserve the characteristics of the reference person. This module is especially suitable for handling cross-category try-on task, such as long sleeves ↔ short sleeves or long pants ↔ skirts, etc., where the exposed arms or legs (including their skin colors and details) can be reasonably predicted. This is good to help the try-on system for formulating a realistic result in the following modules; b) Improved Geometric Matching Module, which is designed to warp clothes according to the geometry of the reference person. By carefully analyzing the basic concept of Thin-Plate Spline (TPS) Duchon (1977) based methods, we argue that the selection of radial basis function is a key point to affect the performance of image warping. Motivated by this end, we then have adopted Wendland's Ψ-function Wendland (1995) as the compactly supported radial basis function. Both theoretical analysis and simulation have verified that the proposed method is able to characterize the local geometrical structure of images, which is good for warping the clothes image, especially with the complex texture; c) Trade-Off Fusion Module, which is to trade off the characteristics of the warped clothes and the reference person, this module is to make the generated try-on images looks more natural and realistic based on a fine-tune symmetry of the network structure (a pair of UNet Ronneberger et al. (2015)). Experiments show that AVTON significantly outperforms the state-of-the-art methods for virtual try-on Han et al. (2018); Wang et al. (2018a); Yu et al. (2019); Yang et al. (2020), and can generate realistic try-on images in all-type clothing try-on task (Fig. 1).

## 2 ARBITRARY VIRTUAL TRY-ON NETWORK

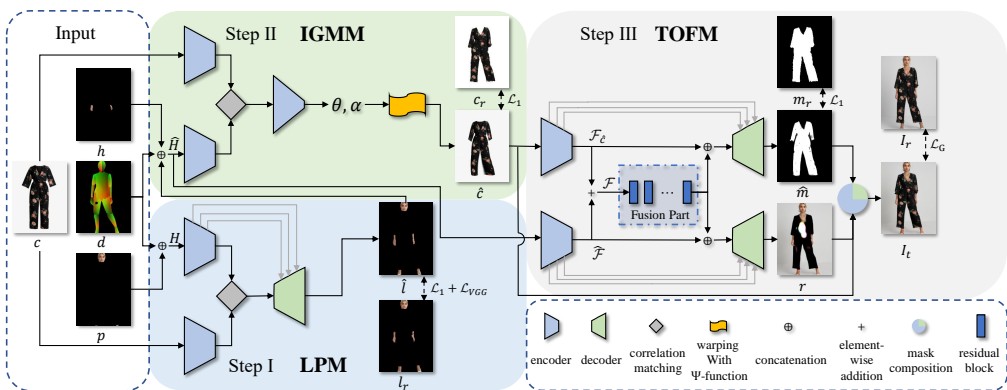

Figure 2: An overview of our AVTON. **Step I**: Limbs Prediction Module takes the target clothes $c$ and the human body information $H$ as the input to predict the exposed limbs and preserve the primary human body information, and output the predicted human body $\hat{l}$; **Step II**: Improved Geometric Matching Module takes the target clothes $c$ and the refined human body information $\hat{H}$ as input, and output the warped clothes $\hat{c}$; **Step III**: Trade-Off Fusion Module firstly takes the warped clothes $\hat{c}$ and the refined human body information $\hat{H}$ as the input to predict the composition mask $\hat{m}$ and the rendered person $r$, and then compose the outputs with the warped clothes $\hat{c}$ to generate the try-on image $I_t$.

Our goal is to learn an arbitrary virtual try-on model that can be adapt to all-type clothing and cross-category try-on tasks and generate more realistic try-on images than prior arts. The proposed AVTON contains three modules, as shown in Fig. 2. First, the limbs Prediction Module (LPM) is utilized to predict the limbs, head, and non-target human body. This is especially useful for

handling cross-category try-on task. Second, the Improved Geometric Matching Module (IGMM) uses Wendland's $\Psi$-function Wendland (1995) to improve the TPS based method Duchon (1977) for warping the clothes. Third, the Trade-Off Fusion Module (TOFM) takes the warped clothes and the human body information as input, and then generates composition mask and rendered person by a pair of UNet Ronneberger et al. (2015) and a fusion part. Especially, the TOFM takes advantage of GAN Goodfellow et al. (2020).

## 2.1 LIMBS PREDICTION MODULE (LPM)

The purpose of designing the Limbs Prediction Module (LPM) is to predict the exposed limbs and preserve the primary human body information (i.e., head, hand details, and non-target human body parts). Most earlier methods generate exposed limbs in try-on steps but neglected primary human body information, which may generate unreasonable color of skin and occlusion. This can usually happen when handling cross-category try-on task, i.e. long sleeve $\rightarrow$ short sleeve, or long pant $\rightarrow$ short pant, as some limbs originally covered by the clothes will be exposed. We thereby propose LPM to address these issues.

Given a reference person image $I_r$, LPM takes the target clothes $c$, the human body information (i.e., the pose and shape information $d$ extracted by DensePose Alp Güler et al. (2018), the head and non-target human body parts information $p$ as mentioned in Section ??) as inputs to predict exposed limbs $\hat{l}$. In detail, the target clothes $c$ and the human body information $H = d \oplus p$ ($\oplus$: concatenation) are firstly encoded as the input features. They are then formed as a single tensor by a correlation layer and input to the decoder, which is to calculate the feature correlation by $\mathbf{M} = \hat{\mathbf{M}}_c^\mathsf{T} \hat{\mathbf{M}}_H$, $\hat{\mathbf{M}}_H$ and $\hat{\mathbf{M}}_c$ are the normalized encoder features of body information $H$ and clothes information $c$, respectively. Finally, the exposed limbs are predicted by the decoder. The encoder and correlation layer are similar to CP-VTON's GMM step Wang et al. (2018a), while the human body information $d$'s encoder-decoder layers are similarly to UNet Ronneberger et al. (2015) structure shown in Fig. 2. All this leads to preserve the primary body information.

The LPM is trained under a combination of the pixel-wise L1 loss and VGG perceptual loss between predicted result $\hat{l}$ and ground truth $l_r$, where $l_r$ includes head, non-target human body parts and exposed limbs in the reference person $I_r$:

$$\mathcal{L}_{\text{LPM}} = \lambda_{\text{L1}} \|\hat{l} - l_r\|_1 + \lambda_{\text{vgg}} \mathcal{L}_{\text{VGG}}(\hat{l}, l_r) \tag{1}$$

where

$$\mathcal{L}_{\text{VGG}}(\hat{l}, l_r) = \sum_{i=1}^{5} \lambda_i \|\phi_i(\hat{l}) - \phi_i(l_r)\|_1 \tag{2}$$

is the VGG perceptual loss, where $\lambda_{\text{L1}}$ and $\lambda_{\text{vgg}}$ are the trade-off parameters for two loss terms in Eq. 1, which all set to 1 in our experiments, and $\phi_i(l)$ denotes the feature map of limbs' image $l$ of the $i$-$th$ layer in the visual perception network $\phi$, which is a VGG19 pre-trained on ImageNet.

## 2.2 IMPROVED GEOMETRIC MATCHING MODULE (IGMM)

The old way of warping clothes is based on Thin-Plate Splines (TPS) Duchon (1977) with $r^2 \log r$ as Radial Basis Functions (RBFs). This method yields minimal bending energy properties measured over the whole image. But since it is not a compactly supported RBFs, the deformation will cover the regions where all control points are located. It is advantageous for yielding an overall smooth deformation and preserving geometrical characteristics, but it is problematic when only a small part of the image is desired to be deformed. This will lead to unreasonable deformation when warps clothes. To address this issue,we in this work has adopted $\Psi$-function of Wendland Wendland (1995) as RBFs. As mentioned in Fornefett et al. (1999; 2001), it is a more compactly support for the registration of images so that the bending region can be narrowed down when minimizing the bending energy. Here, we first give its formulation as follows:

$$\psi_{d,k}(r) := I_{(1-r)_+^{\lfloor d/2 \rfloor + k + 1}}^k(r) \tag{3}$$

where

$$(1-r)_+^v = \begin{cases} (1-r)^v & 0 \le r < 1 \\ 0 & r \ge 1 \end{cases},$$

$$I_{\psi(r)} := \int_r^\infty t\psi(t)\mathrm{d}t \quad r \geq 0.$$

The equation also holds for different spatial supports $\alpha$: $\psi_\alpha(r) = \psi(r/\alpha)$. We apply the $\psi_{\alpha,3,1}$-function as RBFs to replace the TPS's RBFs:

$$\psi_{\alpha,3,1}(r) = (1 - r/\alpha)_+^4 (4r/\alpha + 1) \tag{4}$$

where $\alpha$ is a learnable parameter as same as the spatial transformation parameters $\theta$. Inspired by CP-VTON Wang et al. (2018a), we use the same structure to learn these parameters. As shown in Fig. 2, our IGMM firstly extract high-level features of the target clothes $c$ and the refined human body information $\hat{H} = d \oplus \hat{l} \oplus h$ ($h$ represents hand details in reference person $I_r$) respectively. Then a correlation layer to combine two features into as single tensor as input to the regression network that predicts $\theta$ and $\alpha$. Finally, a transformation $\Psi_{\theta,\alpha}$ based on Eq. 4 for warping the target clothes $c$ into the result $\hat{c} = \Psi_{\theta,\alpha}(c)$.

To learn the module above, we make some derivations and experiments to study the size of spatial supports $\alpha$, which shows a significant positive correlation between spatial warping range and $\alpha$, and we concluded that the best $\alpha$ should meet the condition: $\alpha \geq D$, where $D$ is the maximum distance among control points in Delaunay triangles Delaunay et al. (1934). In our experiments, we set $D = \sqrt{a^2 + b^2}$ ($a$ and $b$ is the vertical distance and the horizontal distance among nearest control points, respectively). Consequently, the final $\alpha = \lambda_\alpha \hat{\alpha} + D$, where $\hat{\alpha}$ is the sigmoid's output of the regression network and $\lambda_\alpha$ is set to 6 in our experiments.

To train the module, we conducted it under the pixel-wise L1 loss between the warped clothes $\hat{c}$ and ground truth $c_r$, where $c_r$ is the clothes worn on the reference person in $I_r$:

$$\mathcal{L}_{\text{IGMM}} = \|\hat{c} - c_r\|_1 = \|\Psi_{\theta,\alpha}(c) - c_r\|_1 \tag{5}$$

## 2.3 TRADE-OFF FUSION MODULE (TOFM)

Try-on image synthesis from the warped clothes and the reference person is a many-to-one mapping problem. It aims at not only preserving the characteristics of the warped clothes and the reference person, but also trading off them to make images realistic. One of the common methods Han et al. (2018) is to produce a composition mask for fusing UNet Ronneberger et al. (2015) rendered person with warped clothes and finally to produce a refined result. Although it can refine the course try-on image, it lacks preserving characteristics of the warped clothes. Another common method Wang et al. (2018a) is to utilize an UNet to render a person image and predict a composition mask simultaneously, and then synthesizing the try-on image by fusing the rendered person and the warped clothes via the composition mask. But this way failed to preserve characteristics of reference person on account of using a single UNet structure, this structure prefers to preserve characteristics of the warped clothes. And other methods Yu et al. (2019); Yang et al. (2020) have some problems with trading off characteristics between the warped clothes and the reference person (e.g., artifacts near the neck regions).

In this paper, we adopt a GAN Goodfellow et al. (2020) based method for generating the realistic try-on image. In detail, we formulate the generator $G$ in GAN by a pair of UNet, where the inputs are the warped clothes $\hat{c}$ and refined human body information $\hat{H}$, while the outputs are the generated composition mask $\hat{m}$ and rendered person $r$. However, making the characteristics of the warped clothes and the reference person contribute equally when generating the try-on image is still problematic, as this will cause occlusion and artifact problems. To make images more realistic, we add a fusion part in our try-on module (Fig. 2 residual blocks) for handling the above problems. Specifically, in the proposed module, the warped clothes' features $\mathcal{F}_{\hat{c}}$ and the refined human body features are firstly summed element-wisely, so that the fused features can be obtained, i.e. $\mathcal{F}(\mathcal{F}_{\hat{c}} + \hat{\mathcal{F}} \to \mathcal{F})$. Then the fused features $\mathcal{F}$ are concatenated with $\mathcal{F}_{\hat{c}}$ and $\hat{\mathcal{F}}$, and decoded respectively to get predicted mask $\hat{m}$ and predicted rendered person $r$. Finally, similarly to CP-VTON Wang et al. (2018a), $\hat{c}$ and $r$ are fused together using $\hat{m}$ to synthesize try-on image $I_t$:

$$I_t = \hat{m} \odot \hat{c} + (1 - \hat{m}) \odot r \tag{6}$$

where $\odot$ represents element-wise matrix multiplication. Additionally, we use the multi-scale discriminators $D$ that is similar to pix2pixHD Wang et al. (2018b).

At the training phase, the generator's loss is the combination loss scheme of CP-VTON and pix2pixHD, it includes L1 loss, VGG perceptual loss (Eq. 2) and LSGAN loss:

$$
\begin{aligned}
\mathcal{L}_G = & \lambda_{L1} \|I_t - I_g\|_1 + \lambda_{vgg} \mathcal{L}_{\text{VGG}}(I_t, I_g) + \\
& \lambda_{mask} \|\hat{m} - m_r\|_1 + \lambda_{lsgan}(D(\hat{c}, \hat{H}, I_t) - 1)^2
\end{aligned}
\tag{7}
$$

while the discirminator's loss is:

$$
\mathcal{L}_D = ((D(\hat{c}, \hat{H}, I_t))^2 + (D(\hat{c}, \hat{H}, I_g) - 1)^2)/2
\tag{8}
$$

where $I_g$ is the ground truth image, and $I_g = I_r$ in training stage, $m_r$ is the mask of $c_r$. In our experiments, we set $\lambda_{L1}$, $\lambda_{vgg}$ and $\lambda_{mask}$ to 10, while set $\lambda_{lsgan}$ to 1.

## 3 EXPERIMENTS

### 3.1 DATASET DESCRIPTION

In this work, we evaluate the performance of our proposed work based on two datasets: the VITON dataset Han et al. (2018) that is used in VITON Han et al. (2018), CP-VTON Wang et al. (2018a) and ACGPN Yang et al. (2020) et al., and the newly-collected Zalando dataset. In this paper we call them VITON-Dataset and Zalando-Dataset respectively.

The VITON-Dataset contains 16253 frontal-view woman and top clothing image pairs, which is split into a training set and a testing set with 14221 and 2032 pairs respectively. We also use the strategy in ACGPN Yang et al. (2020) to score the complexities of images in dataset and divide them into Easy, Medium, and Hard levels, which is used to further evaluate the proposed method and other state-of-the-art methods for handling different levels of try-on task.

Though VITON-Dataset only contains top clothes while lacks of bottom and whole clothes. It cannot be utilized to train the model for handling arbitrary try-on tasks. We thereby in this subsection introduce the newly-collected Zalando-Dataset. The Zalando-Dataset is crawled from https://www.zalando.co.uk/. It contains 34928 frontal-view human (include man and woman) and clothing (include top, bottom, and whole) image pairs. In our study, we split it into training set and testing set with 32746 and 2182 image pairs, respectively. In detail, the training set contains 19185 tops, 10587 bottoms, and 2974 whole clothes, while the testing set contains 1310 tops, 692 bottoms, and 180 whole clothes.

### 3.2 IMPLEMENTATION DETAILS

The experiments are conducted on the VITON-Dataset and Zalando-Dataset respectively, and the results are entirely independent of each other. Followed by steps in Fig. 2, we first train the LPM and then use the LPM's trained results to train the IGMM, followed by training the TOFM with the trained results from LPM and IGMM. On the VITON-Dataset training setup, each module is trained for 400K steps with batch size 4, while on the Zalando-Dataset training setup, each module is trained for 800K steps with batch size 4. Both training setups use the Adam optimizer with $\beta_1 = 0.5$ and $\beta_2 = 0.999$, and the learning rate is fixed at 0.0001. Additionally, the resolution for all input and output images is $256 \times 192$, and we use a single NVIDIA 2080Ti GPU in our experiments; we also use the same steps as the training stage to test the modules. The qualitative results in the easy, medium, and hard cases respectively in the Zalando-Dataset testing stage can be seen in Fig. 4 and Fig. 5.

### 3.3 QUALITATIVE RESULTS

**On the VITON-Dataset**. Fig. 3 shows visual comparisons of our proposed method with VITON Han et al. (2018), CP-VTON Wang et al. (2018a), VTNFP Yu et al. (2019) and ACGPN Yang et al. (2020). To save a lot of work on reproducing them (VTNFP has no official code), we refer to the results from the paper of ACGPN. Note that we validate the official code of ACGPN qualitatively and quantitatively and receive the same results as with ACGPN.

In comparison to VITON and CP-VTON, VTNFP preserves more characteristics by using segmentation representation to preserve the non-target parts, but it does not contain enough details. This

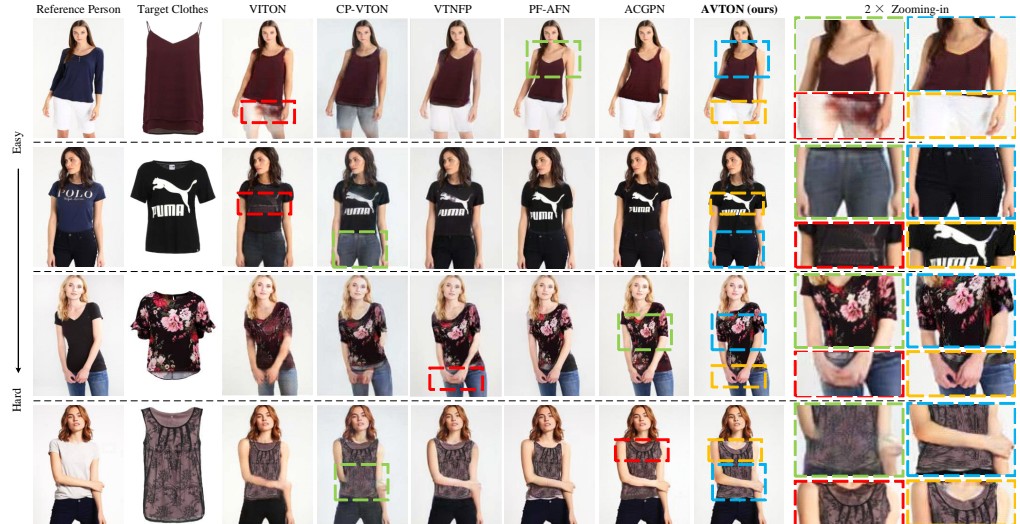

Figure 3: **On the VITON-Dataset**. Qualitative comparisons of VITON Han et al. (2018), CP-VTON Wang et al. (2018a), VTNFP Yu et al. (2019), ACGPN Yang et al. (2020) and AVTON in easy to hard levels (from top to bottom). Our method preserves more characteristics of the reference person with the LPM, and it also preserves more characteristics of the target clothes with the IGMM. What's more, AVTON can generate more realistic try-on images with the TOFM, which is good at trading off characteristics of the warped clothes and the reference person.

happens because of an unawareness of the semantic layout and the relationship within the layout. ACGPN performs better than VTNFP, in that it can preserve hand details, but it also fails to generate try-on details. This is because ACGPN uses TPS to warp clothes (mentioned in Section 2.2), and it uses a simple UNet to fuse the features, which makes it difficult to trade off characteristics of the target clothes and the reference person (mentioned in Section 2.3).

However, AVTON does better both in preserving characteristics and trading off characteristics. Benefiting from the LPM, it predicts limbs first and then provides limb information to the TOFM, which helps solve occlusion problems(e.g., the blue box on the fourth row, the arm is clearer than others). What's more, the IGMM warps clothes more reasonably to preserve styles and patterns (e.g., the blue box on the third row, sleeves are the same length and patterns are clear), and the TOFM makes the try-on image more realistic due to trade-off (e.g., the yellow box on the fourth row, the inner collar should be ignored). In a nutshell, AVTON can generate more realistic try-on images than VITON, CP-VTON, VTNFP, and ACGPN.

**On the Zalando-Dataset**. For fair comparisons, we retrain CP-VTON, Outfit-VITON Neuberger et al. (2020) and ACGPN using the Zalando-Dataset and select the best-trained models. We put on different types of clothes for a person (Fig. 4) and put on one type of clothes for persons of different complexity levels (Fig. 5). It is evident from the results that CP-VTON is not suitable for the all-type clothing try-on task, as CP-VTON uses TPS to warp clothes that we motioned in Section 2.2 and uses a single UNet to generate limbs in the try-on step that we motioned in Section 2.1. Outfit-VITON performs better than CP-VTON, but the content of the generated images are incomplete (the first row in Fig. 4). This can be reasonable because the Outfit-VITON first decouples the features of the clothes and human body, and then decodes them to generate the try-on image. This procedure does not consider the preservation of spatial information, making the image incomplete. In comparison to CP-VTON and Outfit-VITON, ACGPN generates more realistic try-on results, but there are still some issues mentioned in the VITON-Dataset. However, AVTON can deal with these issues by allowing the LPM to accurately predict the color of skin, the IGMM to warp clothes accurately, and the TOFM to make results more realistic.

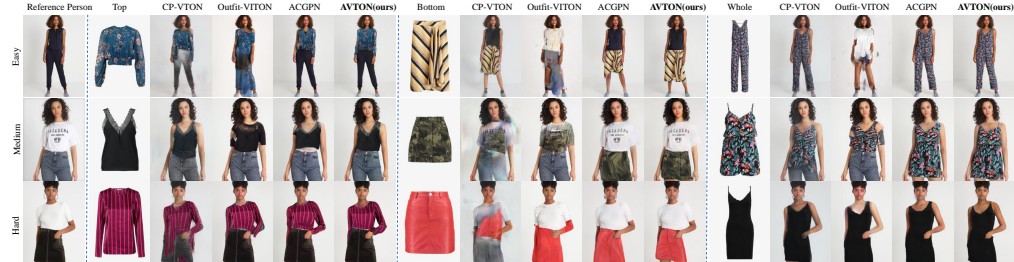

Figure 4: **On the Zalando-Dataset**. Qualitative comparisons of CP-VTON, Outfit-VITON, ACGPN and AVTON with different types of clothing. Our AVTON adapts successfully to the all-type clothing try-on task.

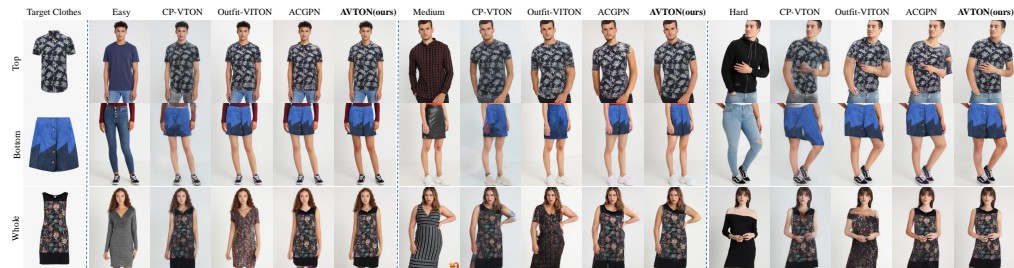

Figure 5: **On the Zalando-Dataset**. Qualitative comparisons of CP-VTON, Outfit-VITON, ACGPN and AVTON at different complexity levels. Our AVTON adapts successfully to cross-category try-on task.

## 3.4 QUANTITATIVE RESULTS

We employ Structure SIMilarity (SSIM) Wang et al. (2004) and Learned Perceptual Image Patch Similarity (LPIPS) Zhang et al. (2018) to measure the similarity between try-on images and groundtruths, and Inception Score (IS) Salimans et al. (2016) to measure the visual quality of try-on images. Specifically, we measure SSIM and LPIPS at different complexity levels on the VITON-Dataset, and measure them with different types of clothes on the Zalando-Dataset.

**On the VITON-Dataset**. TABLE 1 shows quantitative comparisons of our AVTON with VITON, CP-VTON, VTNFP and ACGPN. In our experiments, AVTON obtains a significant lead in all these metrics over baseline methods. Specifically, the SSIM of our method improves by $0.010$, $0.014$, and $0.023$ respectively over that of the best baseline method (i.e., ACGPN) at each complexity level. For LPIPS, our method beats the best baseline method (i.e., ACGPN) by $0.028$, $0.030$, and $0.036$ respectively at each complexity level. And our method surpasses the best baseline method (i.e., ACGPN) by $0.195$ in terms of IS.

| Methods | SSIM↑ / LPIPS↓ | | | | IS↑ |
|---|---|---|---|---|---|
| | Mean | Easy | Medium | Hard | |
| VITON | 0.783 / 0.183 | 0.787 / 0.175 | 0.779 / 0.185 | 0.779 / 0.199 | 2.650 |
| CP-VTON | 0.745 / 0.238 | 0.753 / 0.227 | 0.742 / 0.243 | 0.729 / 0.261 | 2.757 |
| VTNFP | 0.803 / 0.155 | 0.810 / 0.155 | 0.801 / 0.158 | 0.788 / 0.170 | 2.784 |
| ACGPN | 0.845 / 0.107 | 0.854 / 0.101 | 0.841 / 0.110 | 0.828 / 0.119 | 2.829 |
| PF-AFN | 0.849 / 0.101 | 0.857 / 0.095 | 0.845 / 0.091 | 0.820 / 0.104 | 2.883 |
| AVTON (Vanilla) | 0.813 / 0.135 | 0.820 / 0.128 | 0.810 / 0.137 | 0.798 / 0.151 | 2.880 |
| AVTON (w/o Ψ) | 0.819 / 0.123 | 0.826 / 0.116 | 0.816 / 0.126 | 0.805 / 0.137 | 2.983 |
| AVTON (w/o LPM) | 0.856 / 0.090 | 0.861 / 0.084 | 0.852 / 0.093 | 0.849 / 0.101 | 2.859 |
| AVTON (Full) | **0.859 / 0.077** | **0.864 / 0.073** | **0.855 / 0.080** | **0.851 / 0.083** | **3.024** |

Table 1: **On the VITON-Dataset**. SSIM and LPIPS are measured at different complexity levels. AVTON (Vanilla), AVTON (w/o Ψ) and AVTON (w/o LPM) are used for ablation studies.

**On the Zalando-Dataset**. As shown in TABLE 2, we present quantitative comparisons of our AVTON with CP-VTON, Outfit-VITON, and ACGPN, in which CP-VTON, Outfit-VITON and ACGPN are retrained as mentioned in Section 3.3. The SSIM of our method improves by $0.007$,

| Methods | SSIM↑ / LPIPS↓ | | | | IS↑ |
|---|---|---|---|---|---|
| | Mean | Top | Bottom | Whole | |
| CP-VTON | 0.758 / 0.182 | 0.732 / 0.195 | 0.792 / 0.170 | 0.812 / 0.131 | 3.556 |
| Outfit-VITON | 0.787 / 0.155 | 0.763 / 0.172 | 0.831 / 0.127 | 0.792 / 0.137 | 3.722 |
| ACGPN | 0.808 / 0.134 | 0.778 / 0.151 | 0.860 / 0.105 | 0.831 / 0.118 | 3.877 |
| PF-AFN | 0.810 / 0.128 | 0.779 / 0.149 | 0.865 / 0.096 | 0.829 / 0.124 | 3.882 |
| AVTON (Vanilla) | 0.809 / 0.128 | 0.775 / 0.152 | 0.868 / 0.086 | 0.824 / 0.113 | 3.967 |
| AVTON (w/o Ψ) | 0.811 / 0.126 | 0.776 / 0.151 | 0.876 / 0.081 | 0.818 / 0.120 | 3.971 |
| AVTON (w/o LPM) | 0.813 / 0.120 | 0.779 / 0.143 | 0.874 / 0.080 | 0.827 / 0.107 | **4.023** |
| AVTON (Full) | **0.819 / 0.115** | **0.785 / 0.137** | **0.880 / 0.075** | **0.832 / 0.103** | 3.976 |

Table 2: **On the Zalando-Dataset**. SSIM and LPIPS are measured with different types of clothes. AVTON (Vanilla), AVTON (w/o Ψ) and AVTON (w/o LPM) are used for ablation studies.

0.020, and 0.001 respectively over that of the best baseline method (i.e., ACGPN) for each type of clothing. For LPIPS, our method beats the best baseline method (i.e., ACGPN) by 0.014, 0.030, and 0.015 respectively for each type of clothing. And our method surpasses the best baseline method (i.e., ACGPN) by 0.099 in terms of IS.

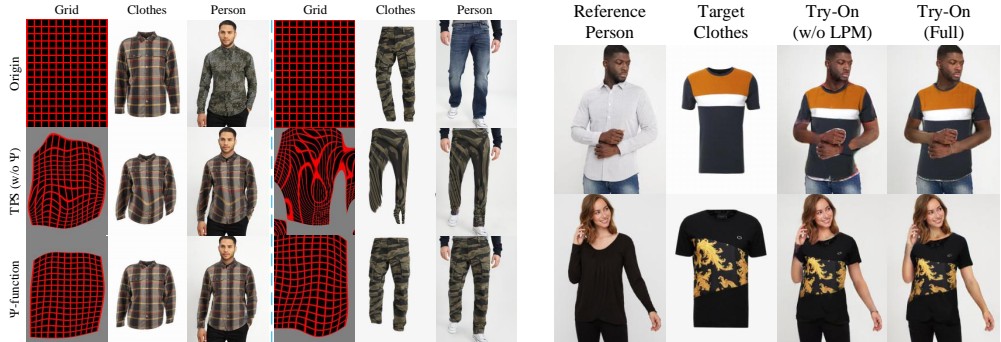

Figure 6: Comparison of TPS and Wendland's Ψ-function.

Figure 7: Comparison of the results with and without the LPM.

## 3.5 ABLATION STUDY

Similarly to the quantitative comparisons, we evaluate the effectiveness of both the LPM and Ψ (Wendland's Ψ-function) using SSIM, LPIPS and IS. As shown in TABLE 1 and TABLE 2, Wendland's Ψ-function plays an important role, where AVTON (Full) surpasses the AVTON (w/o Ψ) by 0.046 and 0.008 in terms of the mean of SSIM, respectively. Here we show the visual comparison in Fig. 6, where Wendland's Ψ-function can warp clothes locally and smoothly, while TPS warps clothes globally. Additionally, the LPM has a significant impact on LPIPS and IS. The LPIPS of AVTON (Full) is reduced by 0.013 (TABLE 1) and the IS of AVTION (Full) is increased by 0.165 (TABLE 1). As shown in Fig. 7, the results (w/o LPM) have broken arms. Note that in TABLE 2, the IS of AVTON (Full) is lower than that of AVTON (w/o LPM). It can be explained that the testing set contains Bottom and Whole cases, most of which do not have occlusion problems. Therefore, the prediction error caused by LPM can be avoided, and the IS of the results without the LPM is higher. In summary, the LPM is necessary for the cross-category try-on task in our experiments.

## 4 CONCLUSION

Deep learning based virtual try-on system has achieved some encouraging progress recently, but there still remain several big challenges that need to be solved, such as trying on arbitrary clothes of all types, trying on the clothes from one category to another and generating image-realistic results with few artifacts. In this paper, we collect a new dataset to enhance the robustness and adaptiveness of the virtual try-on model. Based on this dataset, we then propose a novel virtual try-on network for handling all-type clothing try-on task (tops, bottoms, and whole clothes) and cross-category try-on task (e.g., long sleeves ↔ short sleeves or long pants ↔ skirts, etc.). Extensive simulations and ablation study are conducted. Simulation results based on Quantitative, qualitative evaluation and user study illustrate the great superiority of our AVTON over the state-of-the-art methods.

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

# APPENDIX

## A  THE DETAILED ANALYSIS OF THE COLLECTED ZALANDO-DATASET

In order to show the superiority of the collected Zalando-Dataset, we first illustrate the categories of the dataset, which can be shown in Fig. 1. In detail, the division of dataset based on the categories and genders is shown in Fig. 1(a). From Fig. 1(a), we can see that the collected try-on dataset has involved almost all categories of clothes including both top and bottom clothes for man as well as top, bottom and whole clothes for woman. Fig. 1(b) further show the statistics and distribution of all categories of clothes according to gender and tyoes, where the images of top clothes take up 58.68% while those of bottom and whole clothes take up the remaining 41.32%. Since the conventional VITON-Dataset only involves images with top clothes while lacks of those of bottom and whole clothes, the collected Zalando-Dataset is an extension to VITON-Dataset by involving more clothing categories, which is good for handling real-world arbitrary try-on task.

In order to further show the superiority, we then analyze the characteristics of dataset and compare with those of VITON-Dataset. Here, we choose two measurements for comparisons: 1) the size of the dressed clothes taking up the whole referenced image, and 2) the complexity scores for the referenced image (only for top clothes), which are firstly defined in ACGPN Yang et al. (2020) and are divided into easy, medium and hard cases. The characteristics analysis of proposed dataset and VITON-Dataset based on two measures are shown in Fig. 2.

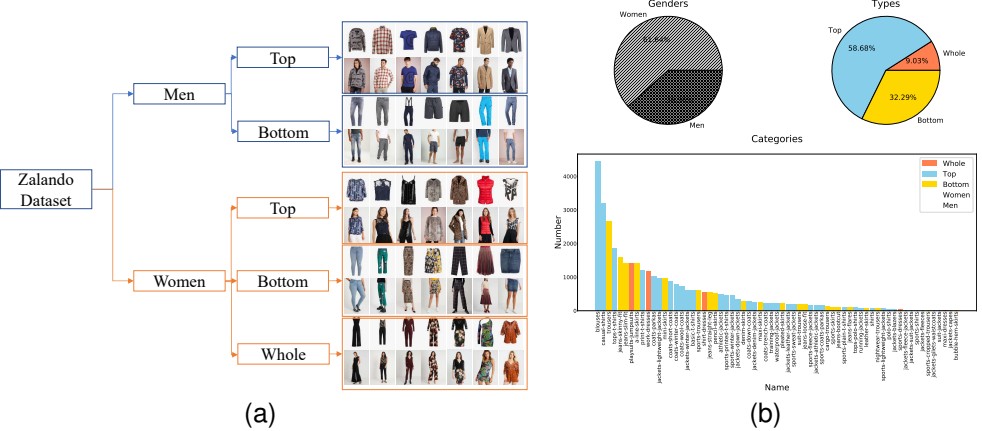

Figure 1: The category division and statistics of Zalando-Dataset: (a) division of dataset; (b) statistics and distribution of dataset.

From Fig. 2(a), we can see that the size of the dressed clothes taking up the whole referenced image has wide range from 0.2 to 0.6 in the collected dataset. While for VITON-Dataset, such range mainly focus on about 0.45. This means that the collected dataset has different sizes of the dressed clothes and is more extensive in data selection. This can also be more realistic and closer to the real-world cases. As a result, a try-on method that can handling different size of the dressed clothes is more useful and practical, which is good to satisfy the requirement of user. From Fig. 2(b), we can see that the distribution of the score curve of the collected dataset is on the left of that of VITON-Dataset. This indicates that the complexity of the proposed dataset is higher than that of VITON-Dataset. Therefore, a method trained on such dataset can handle more complex try-on task such as limb intersections and torso occlusions, which is good to enhance the robustness and adaptiveness of the model.

## B  ANALYSIS OF $\psi$-FUNCTION OF WENDLAND

The $\psi$-function of Wendland utilized in our work is to replace the conventional Radial basis function (RBF) in Thin-Plate Splines (TPS) Duchon (1977) for handling virtual try-on task, and we admit

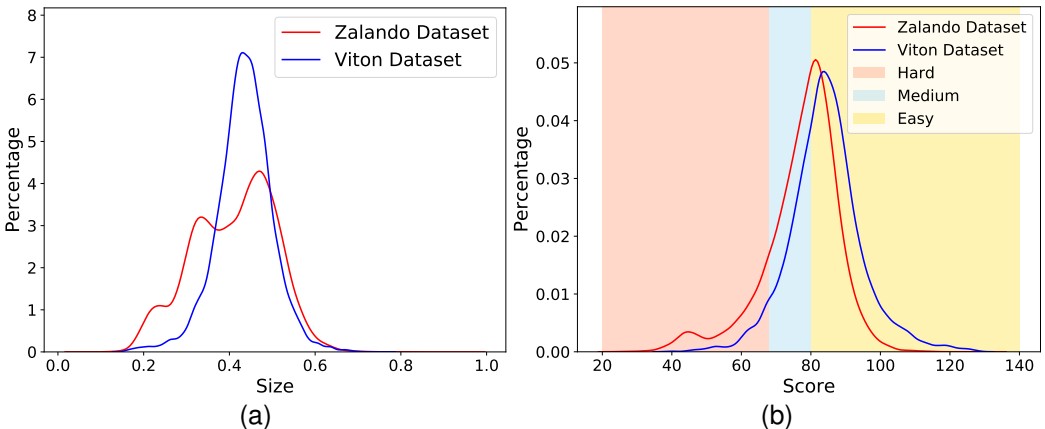

Figure 2: (a) the distribution of the size of the dressed clothes taking up the whole referenced image; (b) the distribution of the complexity scores for the referenced image (only for top clothes).

that it can indeed enhance the performance of try-on task. The reason for the improvement is that $\psi$-function of Wendland is compactly supported for local deformation due to its locality, solvability and efficiency. Here, we will explain it by two issues: **1) why conventional RBF in TPS cannot work well compared with the $\psi$-function of Wendland; 2) why we choose $\psi_{3,1}$-function of Wendland in our work.**

First, we will say the key technique of clothes warping for handling try-on task is Thin-Plate Splines (TPS) Duchon (1977); Fornefett et al. (2001), which has been extensively utilized in VITON Han et al. (2018), CP-VITON Wang et al. (2018), ACGPN Yang et al. (2020) and other state-of-the-art methods. In detail, denote $x$ be the source point in the tiled clothes, $y$ is the corresponding target point in the warped clothes, the goal of TPS is to fit a nonlinear mapping function $f(x)$ between the source points $x$ and the target warped points $y$ by minimizing the following energy function:

$$E_{tps}(f) = \sum_{i=1}^{K} ||y_i - f(x_i)||^2. \tag{1}$$

To solve the above problem, Radial basis function (RBF) is typically involved since it has a natural representation of $f(x)$ for detailed introduction). To calculate the optimal $f(x)$, a set of controlled points in the original space $\{c_i, i = 1, 2, \ldots, K\}$ (the locations of which have already been fixed both in the original space and warped space) should be given. Then, RBF is to defines $f(x)$ as follows:

$$f(x) = \sum_{i=1}^{K} w_i \varphi(\| x - c_i \|) \tag{2}$$

where $\|\cdot\|$ is the usual Euclidean norm, $\varphi(r)$ is the Radial basis kernel in the formulation of $\varphi(r) = r^2 \log r$, $w_i$ is a set of mapping coefficients to be learned. The optimal $f(x)$ can be calculated by a close form solution via nonlinear regression.

Here, as mentioned in Fornefett et al. (2001), the RBF is good for yielding an overall smooth deformation and preserving geometrical characteristics, but it is not compactly supported for local deformation. In detail, following the formulation of Radial basis kernel $\varphi(r) = r^2 \log r$, we can see the further $x$ is from $c_i$, the larger $r = \| x - c_i \|$ is, then the larger Radial basis kernel $\varphi(r) = r^2 \log r$ is (which can be shown as Fig. 3). In such case, $f(x)$ in Eq. equation 2 tends to be very large causing that $x$ will be unnecessarily deformed (we will show an example as follows). On the other hand, a good $f(x)$ should be compactly supported, i.e., $f(x)$ is small given $x$ is far away from $c_i$ so that the far points will not be affected by deformation. This is good for the case that small part of image is desired to be deformed, especially for handling the try-on task that the clothes part in the image is with small size. To the contrast, $\psi_{3,k}$-function of Wendland is compactly supported, which can be shown as Fig. 3.

In order to further show why $\psi_{\alpha,3,1}$-function of Wendland is better RBF based TPS, we will conduct a visual example for comparison, which can be shown as Fig. 4.

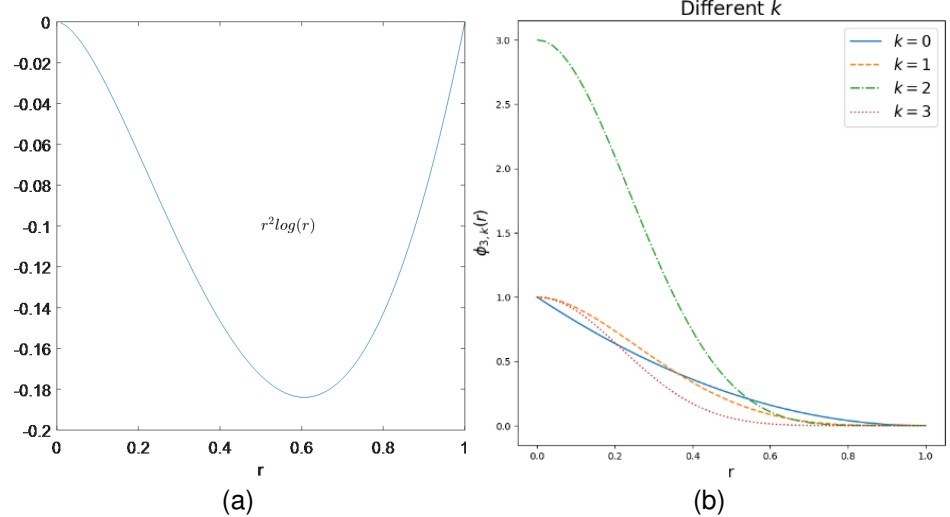

(a)

(b)

Figure 3: The curve of RBF in TPS and $\psi_{3,k}$-function of Wendland with varied $r$: (a) $\varphi(r) = r^2 \log r$ (b) $\psi_{3,k}$-function of Wendland with different $k$

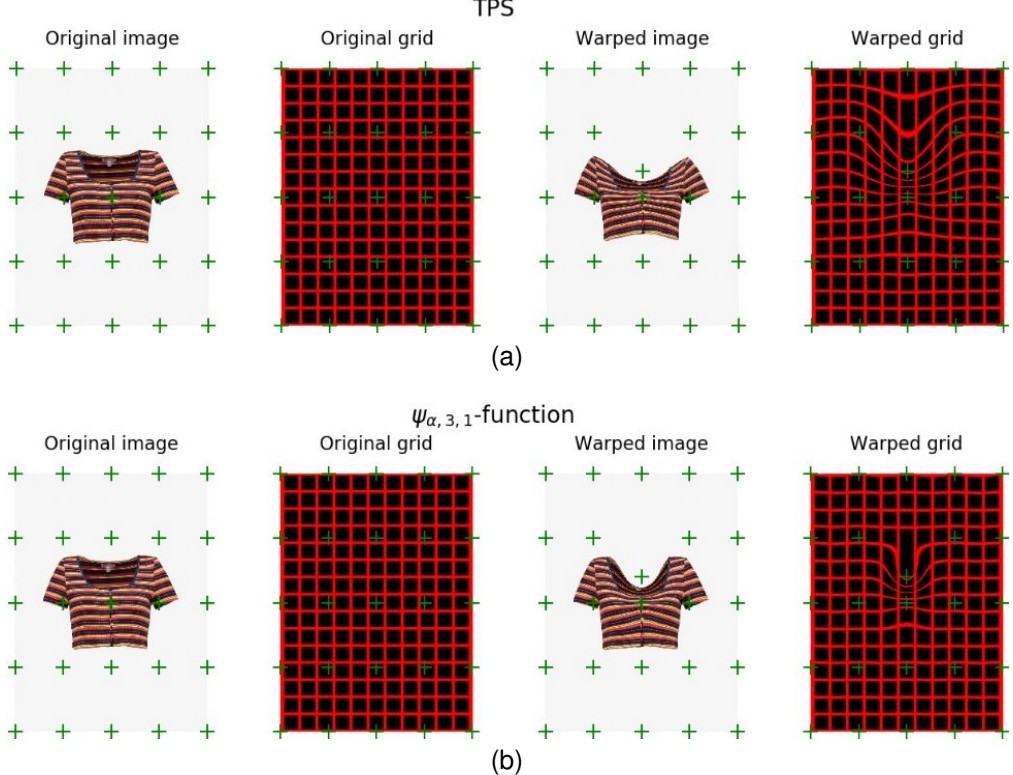

Figure 4: the warped image and gird learned by the TPS and $\psi_{\alpha,3,1}$-function of Wendland: (a) TPS; (b) $\psi_{\alpha,3,1}$-function of Wendland

The first and third columns in Fig. 4 represent the original and warped images, respectively, while the second and the fourth columns represent the original and warped grid, respectively, and the green "+" represents the controlled points both in the original space and warped space. From the above simulation results, we can see that by fixing the same controlled points, the warped grid learned by TPS is more deformed than that learned by $\psi_{\alpha,3,1}$-function of Wendland. This can be obviously observed that the second and third lines in the top grids of TPS has been deformed while those of $\psi_{\alpha,3,1}$-function of Wendland do not make any change. This has verified that the $\psi_{\alpha,3,1}$-function of Wendland is better than RBF based TPS, which is good for handling the try-on task that the clothes part in the image is with small size.

## C  PARAMETER SELECTION FOR $\psi_{3,1}$-FUNCTION OF WENDLAND

We next explain why we choose $\psi_{3,1}$-function of Wendland in our work. The $\psi$-function of Wendland is firstly developed for biomedical image registration Fornefett et al. (2001), which is adopted to replace RBF due to its locality, solvability and efficiency. Its general formulations are as follows:

$$\psi(r) = \begin{cases} p(r) & 0 \le r < 1 \\ 0 & r \ge 1 \end{cases} \tag{3}$$

where $p(r)$ is a univariately polynomial. Let $\psi(r)$ denote the univariate function, then $\psi : \mathbb{R}^d \to \mathbb{R}$, $\psi(||r||) = \psi(||r||)$ is the corresponding multivariate function in the space of dimension $d$. The mathematical property of positive definiteness of $\psi$ depends on the space dimension $d$. If $\psi$ is positive definite on $\mathbb{R}^d$, then $\psi$ is also positive definite on $\mathbb{R}^g$ with $0 < g \le d$. It has been proven in [e] that for given space dimension $d$ and smoothness $C^{2k}(\mathbb{R})$ there exists –up to a constant factor– only one function $\psi(r)$ of the above formulation which is positive definite on $\mathbb{R}^d$ and has a polynomial of minimal degree $\lfloor d/2 \rfloor + 3k + 1$, where $\lfloor x \rfloor$ is the floor function returning the largest integer of $x$. We neglect some math derivation in Fornefett et al. (2001) and give the final function as follows:

$$\psi_{d,k}(r) := I^k_{(1-r)^{\lfloor d/2 \rfloor + k + 1}_+}(r) \tag{4}$$

with

$$(1-r)^v_+ = \begin{cases} (1-r)^v & 0 \le r < 1 \\ 0 & r \ge 1 \end{cases} \tag{5}$$

as the truncated polynomial and

$$I_{\psi(r)} := \int_r^\infty t\psi(t)\mathrm{d}t \quad r \ge 0 \tag{6}$$

as the integral operator.

Following the above analysis, we can see that there are two key parameters in $\psi$-function of Wendland: 1) $d$: the space dimension to guarantee $\psi_{d,k}$ positive definite; 2) $k$: the times for performing integral operator. Here, the dimension is 2 and the floor function $\lfloor d/2 \rfloor$ is equal no matter $d = 3$ or $d = 2$, we will choose $d = 3$ for simplicity. Then, by fixing $d = 3$, we can list $\psi_{3,k}$-function with different $k$, which can be shown as follows:

$$\begin{aligned}
\psi_{30}(r) &= (1-r)^2_+ \\
\psi_{3,1}(r) &= (1-r)^4_+(4r+1) \\
\psi_{3,2}(r) &= (1-r)^6_+(35r^2 + 18r + 3) \\
\psi_{3,3}(r) &= (1-r)^8_+(32r^3 + 25r^2 + 8r + 1)
\end{aligned} \tag{7}$$

To further compare the deformation performance with different $k$, we will also show the deformation grid by fixing some controlled points as in Fig. 5. From Fig. 5, the larger $k$ is, the smoother the deformation grid is. But when $k > 1$, the polynomial will become more complex while their deformations are relatively smaller, we thereby choose $k = 1$. Finally, as analyzed above, we will finally determine $d = 3$ and $k = 1$.

For the choice of the parameter $\alpha$ in $\psi_{\alpha,3,1}(r)$ of Eq. equation 4, we first select 25 controlled points and use Delaunay triangulation to analyze the distances between the control points, as shown in the Fig. 6.

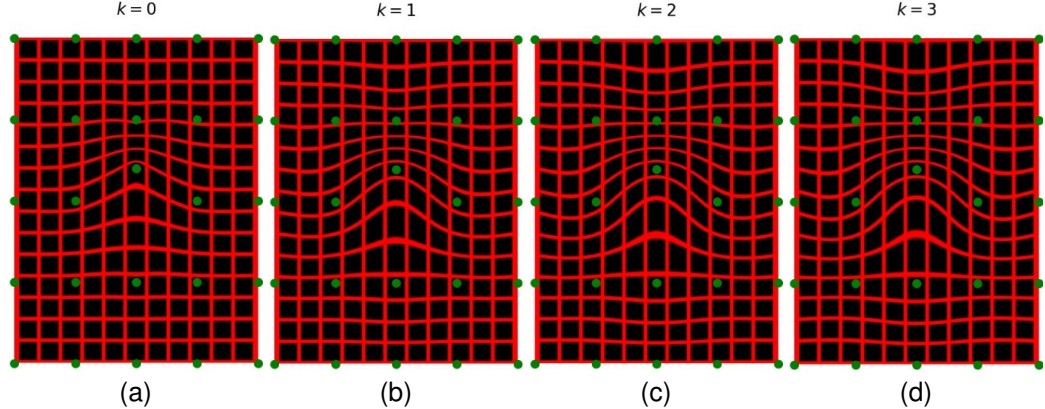

Figure 5: the deformation grid learned with different $k$

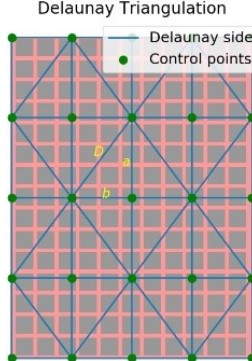

Figure 6: Delaunay triangulation for analyzing the distances of the control points

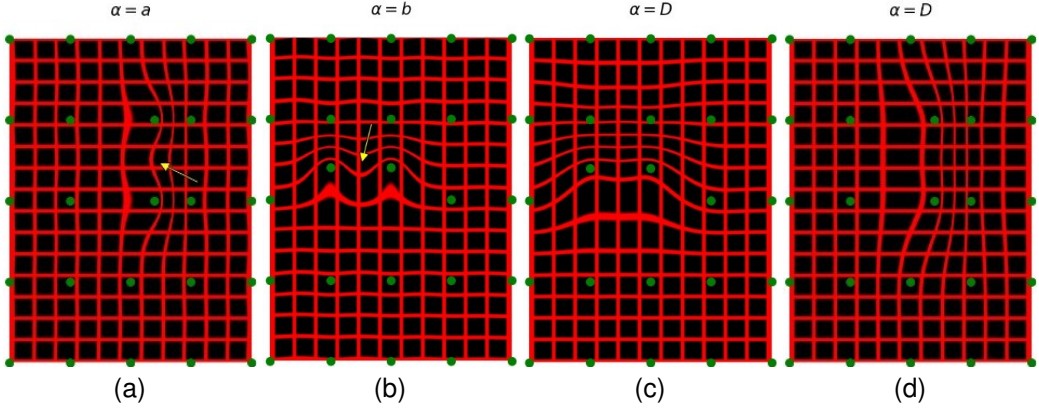

Figure 7: the deformation grid learned with different $\alpha$

In the Delaunay triangle, the distances between the nearest controlled points are $a, b, D$, where $D = \sqrt{a^2 + b^2}$. In order to ensure the interaction between the controlled points, the value of $\alpha$ must be greater than one of the three nearest neighbor distances, so we separately analyze the three cases when $\alpha = a$, $\alpha = b$, $\alpha = D$, as show in Figure 7. Obviously, when $\alpha = a$ or $\alpha = b$, a saddle point will be formed between two nearest points in the vertical or horizontal direction, and when $\alpha = D$, it is flatter between two nearest points in vertical and horizontal directions. Finally, as analyzed above, we will finally determine $\alpha \geq D$.

## D    ABLATION STUDY OF QUALITATIVE RESULTS

Some more simulations for the comparision between TPS and Wendland's $\Psi$-function on VITON and the collected dataset can be seen in Fig. 8 and Fig. 9.

Here, in order to further show why LPM is good for handling cross-category try-on task (e.g., long sleeves/pants $\leftrightarrow$ short sleeves/pants, etc.), we give another ablation study for analysis, where we aim to show the full LPM can well predict some exposed arms or legs when long sleeves/pants $\leftrightarrow$ short sleeves/pants, etc.) compared with LPM w/o correlation layer, LPM w/o U-Net, or even w/o LPM. The simulation results are shown in Fig. 10 and Fig. 11. From simulation results, we can see that the performances of w/o LPM is the worst, since the features of human body and clothes are not well fused. LPM w/o correlation layer and LPM w/o U-Net can achieve relatively better performance. By meriting from the advantages of both correlation layer and U-Net structure, the full LPM can achieve the best performance of limb prediction, since the correlation layer is good to keep the correlated information of human body and clothes, while U-Net structure can well preserve the detailed information of original human body by utilizing shallow features.

## E    ARBITRARY CLOTHING COLLOCATION

In real life, a single clothing virtual try-on cannot meet people's needs, and clothing collocation can improve people's preference for virtual try-on. Hence, we conduct an additional experiment to show the results of arbitrary clothing collocation (Fig. 12). During the experiment, we first try on tops with our AVTON and got the intermediate results, then try on bottoms based on the intermediate results, and finally get the clothing collocation results. It can be seen from the experimental results that due to the characteristics-preserving function of the LPM and IGMM, the characteristics of the target clothes and the reference person can still be retained after two try-on steps. And benefit from the characteristic trade-off function of TOFM, the final try-on images are natural and realistic.

## F    USER STUDY

As shown in TABLE 1 and TABLE 2, we conduct two user studies on the VITON-Dataset and Zalando-Dataset. Within these two studies, we compare the ACGPN Yang et al. (2020) and our proposed method AVTON on the VITON-Dataset in easy, medium, and hard cases, respectively. And we compare the retrained ACGPN and our porposed method AVTON on the Zalando-Dataset in the top, bottom, and whole cases, respectively. Note that we only choose the ACGPN as the baseline, as it is by far the best baseline of all the VITON-Dataset based methods. Specifically, we invite 40 volunteers to complete the experiment. In each study, each volunteer is assigned 50 image pairs and is asked to select the most realistic image out of two virtual try-on results. Both studies show that the AVTON performs better than other methods in all-type clothing and cross-category try-on tasks.

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

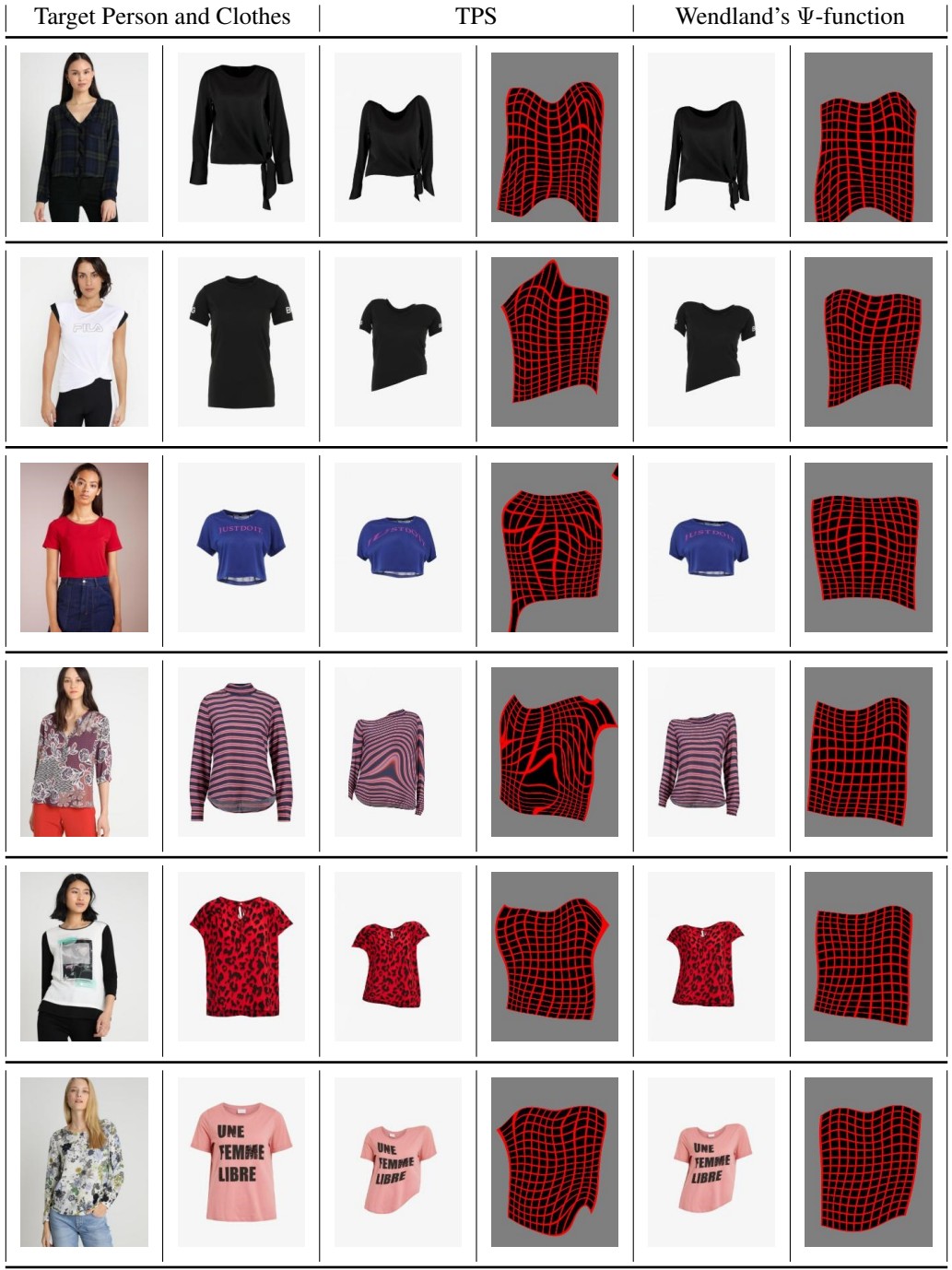

Figure 8: Compasion of image warping between TPS and Wendland's Ψ-function on the VITON dataset

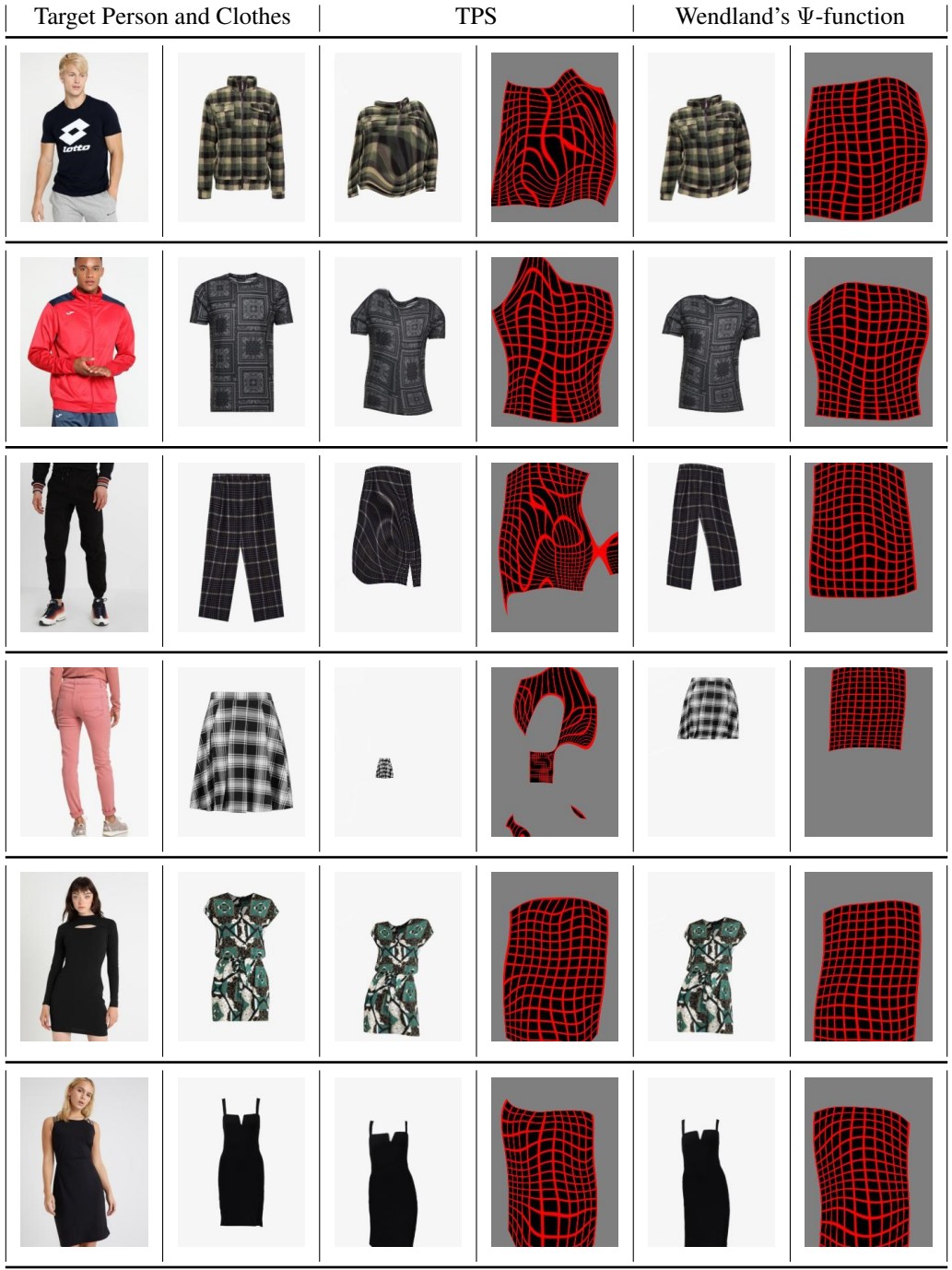

Figure 9: Compasion of image warping between TPS and Wendland's $\Psi$-function on the Zalando dataset

Figure 10: Comparison of the limb prediction results with full LPM, w/o LPM, LPM w/o correlation layer, LPM w/o U-Net structure on the VITON-Dataset

Figure 11: Comparison of the limb prediction results with full LPM, w/o LPM, LPM w/o correlation layer, LPM w/o U-Net structure on the Zalando-Dataset

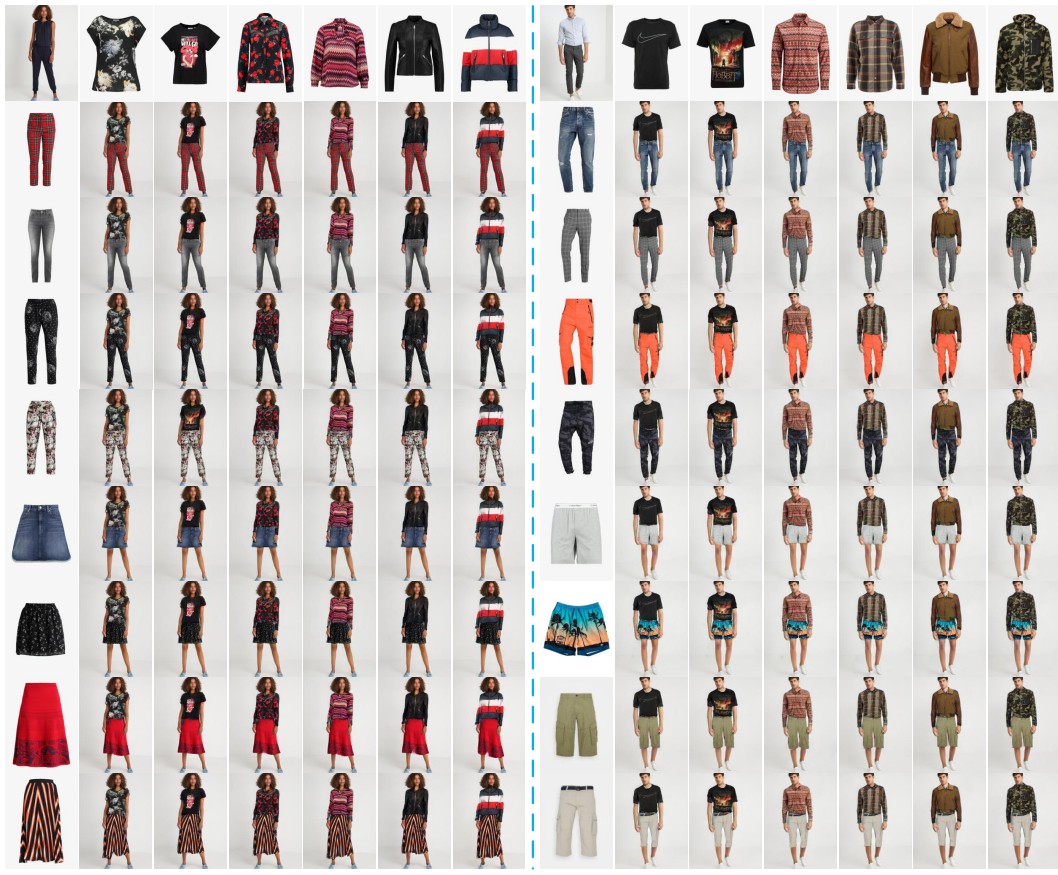

Figure 12: The clothing collocation results in two parts. The upper left image of each part represents the reference person, the upper column of each part represents the target tops, and the left column of each part represents the target bottoms. It can be seen that our AVTON can match clothes arbitrarily.

| Methods | Mean | Easy | Medium | Hard |
|---|---|---|---|---|
| ACGPN | 40.9% | 39.0% | 40.6% | 43.0% |
| AVTON (Full) | **59.1%** | **61.0%** | **59.4%** | **57.0%** |

| Methods | Mean | Top | Bottom | Whole |
|---|---|---|---|---|
| ACGPN | 36.1% | 32.1% | 40.5% | 35.6% |
| AVTON (Full) | **63.9%** | **67.9%** | **59.5%** | **64.4%** |

Table 1: **On the VITON-Dataset**. User study compares ACGPN and our proposed method AVTON at different complexity levels.

Table 2: **On the Zalando-Dataset**. User study compares the retrained ACGPN and our proposed method AVTON with different types of clothes.

Xintong Han, Zuxuan Wu, Zhe Wu, Ruichi Yu, and Larry S Davis. Viton: An image-based virtual try-on network. In *Proceedings of the IEEE conference on computer vision and pattern recognition*, pp. 7543–7552, 2018.

Bochao Wang, Huabin Zheng, Xiaodan Liang, Yimin Chen, Liang Lin, and Meng Yang. Toward characteristic-preserving image-based virtual try-on network. In *Proceedings of the European Conference on Computer Vision (ECCV)*, pp. 589–604, 2018.

Han Yang, Ruimao Zhang, Xiaobao Guo, Wei Liu, Wangmeng Zuo, and Ping Luo. Towards photo-realistic virtual try-on by adaptively generating-preserving image content. In *Proceedings of the IEEE/CVF Conference on Computer Vision and Pattern Recognition*, pp. 7850–7859, 2020.

