# OpenReview forum: "Arbitrary Virtual Try-on Network: Characteristics Representation and Trade-off between Body and Clothing"
_ICLR.cc/2023/Conference — ICLR 2023 poster_

### Official Review · Reviewer_C9dv · 2022-10-22

**Confidence:** 5
**Correctness:** 4
**Technical Novelty And Significance:** 4
**Empirical Novelty And Significance:** 4
**Recommendation:** 8

**Clarity, Quality, Novelty And Reproducibility:**

This paper develops an arbitrary virtual try on network by preserving characteristics representation and trade-off between body and clothes. In my opinion, this work is clearly written and well structured. The novelty of the proposed work is sufficient as the three developed modules are targeted and specific. This work can especially handle the cross-category try-on task, which is the first work I have seen. The structure of the network is also clear making the work reproducible.

**Strength And Weaknesses:**

Strengths:
1. The work is easy to understand and well structured.
2. The novelties of the work are sufficient. The work has developed three modules in the try-on network framework, which are targeted to solve the problems confronted in GAN based 2D virtual try-on task. Specifically, the LPM module is the most important novelty developed in the proposed work. It is to predict the exposed or hidden limbs when perform cross-category try-on task, i.e., long pants<->short pants, this can be the first work pointed out, through which the arbitrary virtual try-on can be realized.
3. Extensive simulations have been conducted to verify the effectiveness of the proposed work both in regular paper and appendix parts. The ablation study and compared work show the superiority of the proposed method.
4. The authors have also collected a new dataset with more categories to train the network for cross-category and arbitrary-category try-on tasks. Based on the dataset, more types of simulations have been conducted to further analyze the proposed method, including arbitrary clothing collocation, user study, qualified analysis on LPM and IGMM.

Weaknesses:
1. The final conclusion is not well written. Some future work can be involved to show how the proposed work can handle real-world application, instead of just writing some sentences from introduction or abstract.
2. The structure of each module can be more detailed. If possible, such structure can be illustrated in the appendix. In addition, some data processing strategy should also be provided.
3. How to get ground truth $c_r$ and $l_r$. Does it need additional annotation? I suggest the authors can explain this point.



**Summary Of The Paper:**

This paper develops an arbitrary virtual try on network by preserving characteristics representation and trade-off between body and clothes. In general, this work is an 2d virtual try-on task, which aims to handle three challengeable issues in GAN based try-on task. How to solve cross-category try-on task, how to well warp the targeted clothes according to the pose and body of targeted human, and how to merge the clothes and body information into the final try-on figures.  After careful reading, the main contributions of the proposed work can be summarized as follows: 1) to develop a new framework with three stages for well solving the above challengeable problems. In detail, the LPM mod	ule is to predict the exposed limbs when long pants are changed to short pants, which tends to be confronted in cross-category try-on task; 2) the IGM module is to designed to warp clothes according to the targeted person based on the locally linear information; 3) trade-off fusion module is to fusion the targeted body and targeted clothes information into final virtual try-on performance by replacing artifact parts. All the three module is targeted and practical. The simulation results including ablation study and comparison work based on new collected dataset are also convincing to verify the effectiveness of the proposed work.


**Summary Of The Review:**

Although there are some issues need to be addressed, based on the quality and novelty of this paper, I believe it can be accepted.

---

> ### Author Response · Authors · 2022-11-13
> **Response to reviewer C9dv**
>
> Thanks for the reviewer’s comments.
>
> 1. Since our collected dataset only contains in-shop images, there are challenges in dealing with images in the wild. So in the future work, we first need to solve the problem of the dataset, and secondly  design a learning strategy to preserve the background content in the wild.
> 2. In the appendix, we introduce the function of the kernel function in detail, but the details of the module are not introduced, mainly considering that the use of the module is based on ResNet and U-Net, these networks are simple and effective. And we will describe the structural details of the modules in detail in the appendix.
> 3. We do not involve any additional annotations to get human parsing parts, such as the head, hand detail, and non-target human body parts. In other words, we definitely not use any human laboring to label the mask of above human body parts to form the ground truth c_r and l_r. Instead, in order to get the correct human parsing parts in our collected Zalando-Dataset, we choose a special human parsing method, namely, Self-Correction-Human-Parsing (SCHP), to get the head, hand detail, and non-target human body parts. In detail, we utilize the inference model of SCHP to directly infer the mask of the human body parts we want.

---

> > ### Comment · Reviewer_C9dv · 2022-11-27
> > **Responses to authors**
> >
> > Thank you for the response to my comments. It is indeed difficult to test a network trained under one dataset distribution to the dataset under another distribution, so I agree with the author's strategy to solve the problem of data set distribution diversity. And given that my concerns regarding $l_r$ and $c_r$ of the claims were all tackled in the rebuttal, I'm happy to recommend acceptance.

---

### Official Review · Reviewer_QoKk · 2022-10-24

**Confidence:** 3
**Correctness:** 3
**Technical Novelty And Significance:** 2
**Empirical Novelty And Significance:** 2
**Recommendation:** 3

**Clarity, Quality, Novelty And Reproducibility:**

The paper is written clearly and each module is explained well.
The novelty of the paper is marginal compared to the recent state of the works.
As a section on how the human parts were generated is missing in the paper I am not sure if their model is reproducible.




**Details Of Ethics Concerns:**

Use of zalando images and crawling Zalando webpage is illegal and introduces copyright issues. The data set can not be shared or used and should be discarded.

**Strength And Weaknesses:**

Strength:
The paper is well written,  is  easy to read and follow, and the descriptions of each module is clear .

The model is evaluated on two different dataset, and it shows improvement compared to VTON, CPVTON, and VTNF.  While showing marginal improvement to ACGPN approach which uses similar ideas as described in this paper.

Their LPM uses a new input type, that represents the person's body better, and does not include clothing items in it, resulting in better final output.

Weakness:
The paper related works are outdated and did not cover several important works which have been done in 2021. For example it would have been good to compare their approach specifically with :

1- VITON-HD: High-Resolution Virtual Try-On via Misalignment-Aware Normalization, Choi, Seunghwan and Park, Sunghyun and Lee, Minsoo and Choo, Jaegul, CVPR 2021,

2- Parser-Free Virtual Try-on via Distilling Appearance Flows, CVPR 2021

3- Dressing in the Wild by Watching Dance Videos, CVPR2022

4- High-Resolution Virtual Try-On with Misalignment and Occlusion-Handled Conditions, Lee, Sangyun and Gu, Gyojung and Park, Sunghyun and Choi, Seunghwan and Choo, Jaegul, ECCV2022

Most of the  work that has been done during the last year covers the same issues raised in this paper and achieved superior results compared to this paper.  For example in 5- ZFlow: Gated Appearance Flow-based Virtual Try-on with 3D Priors, ICCV2021, uses similar representations as input to their model as in LPM.


It is not clear how the segmentation of the body parts(p) in LPM are generated. The reference to the section is missing. I guess the paper has a missing section explaining how the masks are generated?

not sure if their collected dataset from zalando is also anything new as similar dataset was used in VITON-HD: High-Resolution Virtual Try-On via Misalignment-Aware Normalization

Can the model generate a person in pants when only the upper part of the person is given as input?
It would be also good to see the result of the model on diverse clothing, for example a person in tight fitting clothing to a loose dress.

In figure 6 it would be good to compare their cloth deformation module with the work in ACGPN.

It is preferable to report the results with generated image resolution in the tables for better comparisons across different works.




**Summary Of The Paper:**

The paper proposed a method to handle arbitrary clothing in Virtual try on models. The papers argue that recent approaches in virtual try-on lack coverage in cases where the clothing styles change vastly, for example from skirt to pants or long sleeve to short sleeves. To cover such cases they proposed a limb prediction module which takes as input human and target clothing representations and produces the exposed limbs.
Secondly, to cover the clothing deformation better, in their geometric deformation module, they modified previously used Thin-Plate Splines (TPS), by replacing TPS RBF functions with the Phi function of Wendland Wendland. The module takes as input human representations from the Limb prediction module, in addition to clothing and human representations and produces deformed clothing.

Finally, in their Trade-off fusion module (TOFM) they adopt a GAN-based method which uses a pair of UNet as a generator and a multi-scale discriminator D as in pix2pixHD.  The authors argue that to preserve the details of clothing and body parts in the final render image, they use the wrapped clothed features from the geometric deformation module and refined human features ( from LPM) are fused by element-wise sum and concatenated with warped clothed features and human representation features to predict mask and rendered person.




**Summary Of The Review:**

Even though the paper is written clearly, unfortunately it lacks coverage of state of the arts, specifically the works that have been done in 2021, where most of them address the same issues proposed in this work. Comparing this work with last year's works I find the novelty of the paper limited, and the results are not superior.

Some references to a section on how the body parts were generated are also missing.

---

> ### Author Response · Authors · 2022-11-15
> **Response to Reviewer QoKk_part1**
>
> Thanks for the reviewer’s comment.
>
> 1. Regarding to the compared work, we did some compared work with other SOTA methods and summarized as follows:
>
>     - for work 4 and 1, these works mainly focus on generate high-resolution try-on image with or without occlusion condition. We will compare with work 4 for generating try-on image based on different resolutions, i.e., $256\times192$, $512\times384$ and $1024\times768$ (we neglect work 1 as work 4 is an extension to it) based on VITON-HD dataset. The simulation results are given as follows. From simulation results in Table A, we can see that the proposed work can achieve better performance in low-resolution $256\times192$ and $512\times384$, respectively, and achieve competitive performance in 1024*768 compared with HR-VITON. This can be reasonable as the specially designed modules are targeted to handle the try-on work. We will also in the revised manuscript update the results.
>
>         | **Methods**  |  **LPIPS**      | **SSIM**  | **FID**  |  **IS**   |
>         | :------: | :---: | :---: | :--: | :---: |
>         | **HR-VITON** |     0.062      | 0.864 | 9.38 | 2.985 |
>         |  **AVTON**   |     0.055      | 0.872 | 8.91 | 3.01  |
>
>         (a) Comparisons for generating try-on images with $256\times192$ resolution
>
>         | **Methods**  |  **LPIPS**      | **SSIM**  | **FID**  |  **IS**   |
>         | :------: | :---: | :---: | :--: | :---: |
>         | **HR-VITON** |      0.061      | 0.878 | 9.90 | 3.093 |
>         |  **AVTON**   |        0.059      | 0.883 | 9.21 | 3.252 |
>
>         (b) Comparisons for generating try-on images with $512\times384$ resolution
>
>         | **Methods**  |  **LPIPS**      | **SSIM**  | **FID**  |  **IS**   |
>         | :------: | :---: | :---: | :--: | :---: |
>         | **HR-VITON** |        0.065      | 0.892 | 10.91 | 3.142 |
>         |  **AVTON**   |    0.076      | 0.887 | 10.31 | 3.154 |
>
>         (c) Comparisons for generating try-on images with $1024\times768$ resolution
>
>         Table A: HR-VITON v.s. AVTON based on different image generation resolutions. we will add the results as new simulations to the revised manuscript.
>
>     - For work 2, we did some compared work based on VITON and the collected dataset. The simulation are as follows. Simulation results in Table B show that the performance of work 2 is not good as the proposed work in some metrics, e.g., the medium and hard cases in VITON dataset, since in these cases, there are too many occlusions. It also won’t perform better in the collected dataset.
>
>         | DataSets | Methods | SSIM/LPIPS  |     SSIM/LPIPS      |     SSIM/LPIPS      |     SSIM/LPIPS      |  IS   | FID  |
>         | :------: | :-----: | :------: | :------: | :------: | :------: | :---: | :--: |
>         |  **VITON**   |         |    **Mean**     |    **Easy**     |   **Medium**    |    **Hard**     |       |      |
>         |          |  PFAFN  | 0.849/0.101 | 0.857/0.095 | 0.845/0.091 | 0.820/0.104 | 2.883 | 10.09 |
>         |          |  AVTON  | 0.859/0.077 | 0.864/0.073 | 0.855/0.080 | 0.851/0.083 | 3.024 | 9.98 |
>         | **Zalando**  |         |    **Mean**     |     **Top**     |   **Bottom**    |   **Whole**    |       |      |
>         |          |  PFAFN  | 0.810/0.128 | 0.779/0.149 | 0.865/0.096 | 0.829/0.124 | 3.882 | 12.50 |
>         |          |  AVTON  | 0.819/0.115 | 0.785/0.137 | 0.880/0.075 | 0.832/0.103 | 3.976 | 11.88 |
>
>         Table B: Addition simulations by comparing AVTON with PFAFN, we will update the results to Table 1 and 2 in the revised manuscript.
>
>     - for work 3, we indeed want to conduct some results for evaluating the proposed work for video try-on. But we cannot realize it as the group won’t share the dataset for us though we have applied for it many times. But I will say our work can be practically extended for handling video try-on task by inducing optical flow to grasp the temporal consistency between frames. Our future work can lie in such work.
>
>     - for work 5, we have found that its way for calculating the metrics is not consistent with us. But we cannot fairly compare with it since it does not share the code. Therefore, we cannot involve the simulation results in the revised work.
>
> In summary, our work mainly focuses on cross-category and arbitrary try-on task. The collected dataset and specially designed module are all targeted though some certain metrics are not the best in some cases.

---

> > ### Author Response · Authors · 2022-11-15
> > **Response to Reviewer QoKk_part2**
> >
> > 2. Regarding the segmentation of body parts (p) in LPM, we will say in our work, we do not involve any additional annotations to get human parsing parts, such as the head, hand detail, and non-target human body parts. In other words, we definitely not use any human laboring to label the mask of above human body parts to form the ground truth $c_r$ and $l_r$. Therefore, our work is certainly fair for comparisons with previous or state-of-the-art try-on methods.
> >
> >     - In fact, the VITON dataset uses the human parsing method (SSL [a]) to obtain the human parsing parts. Our proposed method and other previous state-of-the-art try-on methods (VITON [b], CP-VTON [c], ACGPN [d], etc.) all use the human parsing parts in VITON-Dataset. Instead, in order to get the correct human parsing parts in our collected Zalando-Dataset, we choose a special human parsing method, namely, Self-Correction-Human-Parsing (SCHP), to get the head, hand detail, and non-target human body parts. In detail, we utilize the inference model of SCHP to directly infer the mask of the human body parts we want. The reason we choose SCHP for human parsing is that it is a noise-tolerant method, which can well handle incorrect labels in ground-truth masks. This can usually happen that since the ambiguous boundary between different semantic parts and those categories with similar appearances are usually confusing for annotators. In addition, SCHP is also model-agnostic and can be applied to any human parsing models for further enhancing their performance. Benefiting from such superiorities, SCHP achieves the new state-of-the-art results on 6 benchmarks and win the 1st place for all human parsing tracks in the 3rd LIP Challenge (human parsing challenge). This is why we choose SCHP as the human parsing methods to form correct human body parts in our work.
> >
> >     [a] K. Gong, X. Liang, X. Shen, and L. Lin. Look into person: Self-supervised structure-sensitive learning and a new benchmark for
> >     human parsing. In CVPR, 2017.
> >
> >   Here, by taking reviewer’s comment into account, we have carefully illustrated this point in revised manuscript.
> >
> > 3. Regarding the details of the Zalando-Dataset dataset, we will say that the Zalando-Dataset is crawled from https://www.zalando.co.uk/, which is a public and famous website providng clear, extensive and large-scale clothes images for fashion analysis. Here, our collected dataset contains 34928 frontal-view human (man and woman) and clothing (top, bottom, and whole) image pairs. In our study, we split it into training and testing set with 32746 and 2182 image pairs, where the training set contains 19185 tops, 10587 bottoms, and 2974 whole clothes, while the testing set contains 1310 tops, 692 bottoms, and 180 whole clothes.
> >
> >     - In addition, we also make some comparison work between the VITON dataset and the Zalando dataset. Firstly, we make the statistics and distributions of all categories of clothes according to gender and styles, where the images of top clothes take up 58.68% while those of bottom and whole clothes take up the remaining 41.32%. Since the conventional VITON-Dataset (available at https://github.com/minar09/ACGPN) only involves images with top clothes while lacks of those of bottom and whole clothes, we will say the collected Zalando-Dataset is an extension to VITON dataset by involving more clothes categories, which is good for handling real-world arbitrary try-on task;
> >
> >     - In order to further show the superiority of the collected dataset, we will analyze the characteristics of the dataset and compare with VITON dataset. Here, we choose two measurements for comparisons: 1) the size of dressed clothes taking up the whole referenced image, and 2) the complexity scores for referenced image (only for TOP), which are firstly defined in ACGPN and are divided into easy, medium and hard cases.
> >
> >         - By analyzing the size, we found that the clothing size in VITON-Dataset is concentrated around 0.4 (which means that the clothing occupies 2/5 of the image). While the clothing size distribution in Zalando-Dataset is between 0.3$\sim$0.5 (that is, the clothing accounts for 3/10$\sim$1/2 of the image). This means that the collected dataset has different sizes of dressed clothes and is more extensive and general in data selection. This can also be more realistic and closer to the real-world applications. As a result, a try-on method that can handling different size of dressed clothes is more useful and practical, which is good to satisfy the requirement of user.
> >
> >         - By analyzing the complexity, we found that the Zalando-Dataset is higher than that of VITON-Dataset. Therefore, a method trained on such dataset can handle more complex try-on task such as Limbs’ intersections and torso occlusions, which is good to enhance the robustness and adaptiveness of the model.
> >
> > Here, by taking reviewer’s comment into account, we will in revised manuscript include some introductions for the collected dataset.

---

> > > ### Author Response · Authors · 2022-11-15
> > > **Response to Reviewer QoKk_part3**
> > >
> > > 4. Regarding the question “Can the model generate a person in pants when only the upper part of the person is given as input?”, we admit there indeed exists such work. A case in point is “FiNet: Compatible and Diverse Fashion Image Inpainting”, where a person in pant can be generated by only giving the upper part of the person while occluding the bottom part. In this work, the model can indeed generalize diverse clothing.
> > >
> > >     - Here, we have to say our work and other related work, such as VITON, CP-VITON, ACGPN etc., mainly focus on one-to-one try-on task, i.e., transferring one targeted clothes to one targeted person to formulate try-on image. This is of great practice as online shopping website usually need to display the try-on performance of a certain clothes on a targeted model. This is quite different from the basic idea of FiNET. However, we will say if we do not provide a targeted pant but provide a random implicit representation of pant without any constraint, we may get some try on image performance but the results certainly cannot be controlled to some extent. Our future work can lie in such work to generate diverse clothing with certain constraint.
> > >
> > > 5. Regarding the comparison with the cloth deformation module with the work in ACGPN, we will say Radial Basis Function (RBF) kernel in Thin-Plate Spline is good for yielding an overall smooth deformation and preserving geometrical characteristics, but it is not compactly supported for local deformation. To solve the problem, ACGPN is to extend the original RBF kernel with a second-order difference constraint on the clothes warping to preserve the local geometrical information.
> > >
> > >     - In our work, we utilize another strategy for characterizing the geometry of image warping by adopting another kernel, i.e., $\Psi$-Function of Wendland to replace RBF kernel and to form a network layer. In fact, the $\Psi$-Function of Wendland is firstly developed for biomedical image registration, and is to replace RBF due to its locality, solvability and efficiency. Both theoretical analysis and simulations in appendix sufficiently show the superiority of $\Psi$-Function of Wendland to RBF. Here, by taking the reviewer’s comments, we indeed compare the image warping performance of $\Psi$-Function of Wendland with the one in ACGPN. The simulation results show that the $\Psi$-Function of Wendland can act on the clothing deformation locally and control the local range. While the constrained TPS (ACGPN) uses the loss function to constrain the adjacent control points and cannot handle the large-scale deformation problem well. However, we cannot update the results to the open-review in current stage, as the images are not allowed to upload to the system. These simulation results will be updated to Fig. 6 in the revised manuscript.
> > >
> > > 6. Regarding the results with generated image resolution in the tables, we will follow the work in VITON-HD for evaluating the try-on performance based on different resolutions, i.e., $256\times192$, $512\times384$ and $1024\times768$, and compare the performance of the proposed work with CP-VTON, ACGPN and VITON-HD. The simulation results are given as Table A as in Response to Reviewer QoKk_part3.
> > >
> > >     - From simulation results, we can see, the proposed work can achieve the best performance in the resolutions $256\times192$ and $512\times384$, respectively, and achieve competitive performance in $1024\times768$ compared with VITON-HD. Here, by taking the reviewer’s comment into account, we will in the revised manuscript update the results.
> > >
> > > We will share the code and datasets in near future.

---

> > > > ### Comment · Reviewer_QoKk · 2022-11-27
> > > > **going through the authors answers I am convinced that paper is not ready for publication**
> > > >
> > > > The paper needs major revision adding all the related work in introduction, related work section and results. I believe that it is job of the authors to look for all the related works before submitting a paper not the reviewer to point it out to them.
> > > > At the moment it looks like cherry picking some related works and putting in the paper.
> > > > Even looking at their comparison  they are either not better than related work or marginally better.
> > > >
> > > > Usage of zalando dataset is illegal and raises copy right issues. The dataset can not be published or use in further research.

---

> > > > > ### Author Response · Authors · 2022-11-29
> > > > > **Rebutal to the Reviewer QoKk**
> > > > >
> > > > > We thank for the reviewer's comment.
> > > > >
> > > > > Regarding to the writing of the manuscript, we will say the main purpose of our work is to develop an arbitrary and cross-category try-on task. This motivation is quite different from current try-on works, and that is why we adopt a new dataset for training the model, and specifically develop the LPM module, the improved image warping module and fusion module for handling cross-category try-on task, improving the image warping and compose try-on performance. The three modules are targeted to handle the challenges of arbitrary try-on task. Our manuscript is therefore written following this motivation. Also, there are quite lots of simulations verify the effectiveness of the proposed methods both in formal paper and appendix.
> > > > >
> > > > > Regarding to the related work, we do not quite agree with the comment of the reviewer, as we indeed have made a detailed survey on this topic and are quite familiar with the SOTA methods the reviewer mentioned (we read related work almost every day from 2d to 3d, from single image to video). In detail, in my opinion, the main timeline of 2D-GAN based try-on methods should be VITON->CP-VITON->AGCPN. That is why almost all methods have been compared with them. In addition, though there are quite lots of other 2D-GAN based methods, they only focus on a certain aspect to improve the performance. For example, the PFAFN has the faster reference time since it adopts the knowledge distillation from a large complex network but its performance has limited improvement; VITON-HD method is proposed to handle high-resolution try-on methods, but its computational complexity is very large and has little superiority in low-resolution try-on task. Every new method has its advantage and disadvantage. Having said that, there currently exists no work focused on arbitrary try-on and our work is to focus on this topic. We believe our work has some contributions including both dataset and specially designed modules. Our work also does coincide with any existing related work.
> > > > >
> > > > > Still about the related work, since there are quite lots of methods, we can only review some main stream works and other related ones due to the very strict page limitations. We cannot leave too much space on related work. Otherwise, the methods and simulations will not have much space to illustrate (though we have already involved many simulations on appendix). By taking the reviewer’s comment, we will review some representative works in revised manuscript.
> > > > >
> > > > > Regarding to the issue of dataset, we will say our work is cooperated with a big IT company and the dataset is provided by the company, where the company mainly response for obtaining the authorization for utilizing the dataset while avoiding illegal risk (also make some cleaning and preprocessing work on the dataset). Having carefully consulted with the company, we guarantee that there is no any illegal problem and we can use it in current research. We will further consider if it is good or not to publish after further consulting.
> > > > >
> > > > > On the other hand, the virtual try-on benchmark dataset VITON and VITON-HD (high resolution version of VITON) are also collected from zalando website, and they have already been published in https://github.com/minar09/ACGPN and https://github.com/shadow2496/VITON-HD, respectively, which are free for everyone to download for research. The two datasets also do not have any illegal issue for utilization of datasets.

---

> > > > > > ### Comment · Reviewer_QoKk · 2022-12-12
> > > > > > **related work out dataed**
> > > > > >
> > > > > > The last related work covered in this paper is from year 2020, you mean we do not have any advances since then? as you also mentioned in above results and the related work I pointed out we could see that the related work of the paper is out dated, hence the novelty of the work also as similar work has been done in year 2021 and 2022.

---

### Official Review · Reviewer_rQV8 · 2022-10-25

**Confidence:** 5
**Correctness:** 4
**Technical Novelty And Significance:** 4
**Empirical Novelty And Significance:** 4
**Recommendation:** 8

**Clarity, Quality, Novelty And Reproducibility:**

The author proposed a deep learning-based virtual try on algorithm. In my opinion, this work is clearly written and well organized. It contains three components. The first component deform the target cloth to the same pose as the human body. The second module takes the human pose and non-target human body parts extracted from image to identify limbs. The third module took deformed cloth and limbs to form a new image. The paper given enough detail to make re-implement their algorithm possible.



**Strength And Weaknesses:**

Strengths:
1.	The collected all-type cloth data set will contribute to other researcher work in the field.
2.	The rich experimental section. Good numerical results in terms of existing metrics (LPIPS, SSIM, IS) and on the user study indicate better performance in the virtual try-on task. Also, the rich qualitative analysis helps appreciating the model qualities.
3.	The proposed system includes modules for limbs prediction, geometric matching, and trade-off fusion. Their limb prediction module can help to preserve body characters for synthesizing cross-category try-on images.
4.	The introduction of the Wendland's Ψ-function in the warping module. To the best of my knowledge, this is the first work to propose to employ this support after the well-established TPS-based warping.
5.	The overview figure is clear enough to help user understand the structure of the system.
Weakness:
Include some failure cases may help to evaluate the limitation of the system.


**Summary Of The Paper:**

This paper proposed a new architecture for virtual try-on and a new method to warp the clothing image. To improve the performance on a variety of different clothes categories (tops, lower-body clothes as skirts and trousers, and dresses) they collect a new dataset. This dataset, called Zalando dataset, is richer than the viton dataset in terms of variety and thus provides a preferable training and testing ground. Based on the dataset, they introduce Wendland functions as RBFs to warp clothes and an Arbitrary Virtual Try-On Network (AVTON) for synthesizing try-on image on all-type clothes. In the experiment, they conduct both qualitative and quantitative comparisons to evaluate their approach against state-of-art methods. Also, a user study is conducted to evaluate the result.



**Summary Of The Review:**

There are some minor issues should be addressed:
1.	The author should provide a more detailed introduction of the dataset.
2.	Include some failure cases may help to evaluate the limitation of the system.
Based on the evaluation of the paper, I will make accept decision for this work.

---

> ### Author Response · Authors · 2022-11-13
> **Response to reviewer rQV8**
>
> Thanks for the reviewer’s comments.
>
> 1. Regarding the content of the dataset, we did some comparison work between the VITON dataset and the Zalando dataset. We made a division of dataset based on the categories and genders that the Zalando-Dataset has involved almost all categories of clothes including both top and bottom clothes for man as well as top, bottom and whole clothes for woman. Furthermore, we made the statistics and distributions of all categories of clothes according to gender and styles, where the images of top clothes take up 58.68% while those of bottom and whole clothes take up the remaining 41.32%. Since the conventional VITON-Dataset only involves images with top clothes while lacks of those of bottom and whole clothes, we will say the collected Zalando-Dataset is an extension to VITON dataset by involving more clothes categories, which is good for handling real-world arbitrary try-on task. In order to further show the superiority of the collected Zalando-Dataset, we will then analyze the characteristics of the dataset and compare with those of VITON dataset. Here, we choose two measurements for comparisons: 1) the size of dressed clothes taking up the whole referenced image, and 2) the complexity scores for referenced image (only for TOP), which are firstly defined in ACGPN and are divided into easy, medium and hard cases.
>     - By analyzing the size, we found that the clothing size in VITON-Dataset is concentrated around 0.4 (which means that the clothing occupies 2/5 of the image). While the clothing size distribution in Zalando-Dataset is between 0.3\~0.5 (that is, the clothing accounts for 3/10\~1/2 of the image). This means that the collected dataset has different sizes of dressed clothes and is more extensive and general in data selection. This can also be more realistic and closer to the real-world applications. As a result, a try-on method that can handling different size of dressed clothes is more useful and practical, which is good to satisfy the requirement of user.
>     - By analyzing the complexity, we found that the Zalando-Dataset is higher than that of VITON-Dataset. Therefore, a method trained on such dataset can handle more complex try-on task such as Limbs’ intersections and torso occlusions, which is good to enhance the robustness and adaptiveness of the model.
>
> 2. We have found that most of the failure cases come from incorrect preprocessing (such as extracting posture through DensePose [2] model and generating human body mask through SCHP [27]). Here, by taking the reviewer’s comment into account, we will especially add a subsection in simulations to include some failure cases for evaluating the limitation of the system.

---

> > ### Comment · Reviewer_rQV8 · 2022-11-28
> > **My reply to the rebuttal**
> >
> > Thank you for the thorough response. The rebuttal addresses my concerns/questions, the dataset collected by the author is richer and more complex than the VITON dataset, and I think the new dataset is more challenging in the virtual try-on task. And for the virtual try-on task, the upstream network (segmentation or detection) does have a certain impact on the subsequent image synthesis, which is also one of the challenges of virtual try-on. In general, faster carefully reading the author’s rebuttal and other people’s comments, I am satisfied with the revised content.

---

### Official Review · Reviewer_NW3C · 2022-10-25

**Confidence:** 3
**Correctness:** 4
**Technical Novelty And Significance:** 3
**Empirical Novelty And Significance:** 2
**Recommendation:** 5

**Clarity, Quality, Novelty And Reproducibility:**

Although there are some writing errors, this paper is well-organized and the overall logic is clear. It changes the radial basis function of TPS and takes it as the main contribution, and extensive analysis contents have proved its effectiveness. However, the architecture basically follows the previous methods, which does not reflect a more significant innovation.

**Strength And Weaknesses:**

Paper Strengths:
1. The functions and principles of the three sub-modules are clearly described and illustrated.
2. This paper makes an improvement in the mathematical formulation of TPS, and there are extensive analysis contents to verify its effectiveness.
3. This paper provides a new dataset, which contains a richer variety of clothes.
Paper Weakness:
1. Compared with the top clothes, what is the main difficulty of the bottom clothes and whole clothes? In addition to the newly collected dataset, whether the algorithm part is specially designed to handle different clothes categories?
2. The dataset collected by this paper is not introduced in detail, such as the number, resolution, and proportion of clothes categories.
3. From 2021 to 2022, there are a lot of works on image-based virtual try-on, however, this method is not compared with them, the latest work compared in this paper is ACGPN in 2020.
4. Most extensive methods use FID to measure the similarity of data distributions between the generated results and the referenced images, which does not appear in this paper.
5. Several grammar and writing errors can be found in the paper.


**Summary Of The Paper:**

This paper proposes a novel virtual try-on network with a newly collected dataset for handling the all-type clothes try-on task and the cross-category try-on task. It consists of three sub-modules: 1) Limbs Prediction Module, which is designed for predicting the human body parts by preserving the characteristics of the reference person; 2) Improved Geometric Matching Module, where the radial basis function of TPS is replaced by the Wendland’s ψ-function to warp clothes more reasonably; 3) Trade-Off Fusion Module, which aims to trade off the characteristics of the warped clothes and the reference person. Extensive experiments are conducted to demonstrate the superior performance of the proposed method.

**Summary Of The Review:**

Virtual try-on has great commercial value and application prospects. In this paper, three sub-modules are designed to handle the all-type clothes try-on task and the cross-category try-on task. It is relatively novel to make an improvement in the mathematical formulation of TPS. However, the architecture basically follows the previous methods, and it is not clear which part is explicitly designed for all-type clothes. Besides, it lacks comparisons with the latest methods in recent two years.

---

> ### Author Response · Authors · 2022-11-14
> **Response to Reviewer NW3C_part2**
>
> We must thank for the reviewer’s comment.
>
> 3. Regarding the comments of compared work, we have involved two representative works for comparison, which are Parser-Free Virtual Try-on via Distilling Appearance Flows (PFAFN), CVPR 2021, High-Resolution Virtual Try-On with Misalignment and Occlusion-Handled Conditions (HR-VITON), ECCV 2022. The first one is to employ knowledge distillation to reduce the dependency of human parsing by predicting the appearance flows, while the second one is to generate high-resolution try-on image in occlusion condition. Here, to be fair, we compare PFAFN with the proposed method based on VITON and the collected dataset, and compare HR-VITON based on VITON-HD dataset (a high-resolution try-on dataset). The simulations are as follows. From the Table A, we can see the proposed work can generally outperform PFAFN in almost all metrices, especially in the medium and hard cases in VITON dataset as there are too many occlusions in these cases. It also won’t perform better in the collected dataset; from Table B, the proposed work can achieve better performance in low-resolution $256\times192$ and $512\times384$, respectively, and achieve competitive performance in $1024\times768$ compared with HR-VITON. This can be reasonable as the specially designed modules are targeted to handle the try-on work. We will also in the revised manuscript update the results.
>
>     | DataSets | Methods | SSIM/LPIPS  |     SSIM/LPIPS      |     SSIM/LPIPS      |     SSIM/LPIPS      |  IS   | FID  |
>     | :------: | :-----: | :------: | :------: | :------: | :------: | :---: | :--: |
>     |  **VITON**   |         |    **Mean**     |    **Easy**     |   **Medium**    |    **Hard**     |       |      |
>     |          |  PFAFN  | 0.849/0.101 | 0.857/0.095 | 0.845/0.091 | 0.820/0.104 | 2.883 | 10.09 |
>     |          |  AVTON  | 0.859/0.077 | 0.864/0.073 | 0.855/0.080 | 0.851/0.083 | 3.024 | 9.98 |
>     | **Zalando**  |         |    **Mean**     |     **Top**     |   **Bottom**    |   **Whole**    |       |      |
>     |          |  PFAFN  | 0.810/0.128 | 0.779/0.149 | 0.865/0.096 | 0.829/0.124 | 3.882 | 12.50 |
>     |          |  AVTON  | 0.819/0.115 | 0.785/0.137 | 0.880/0.075 | 0.832/0.103 | 3.976 | 11.88 |
>
>     Table A: Addition simulations by comparing AVTON with PFAFN, we will update the results to Table 1 and 2 in the revised manuscript.
>
>     | **Methods**  |  **LPIPS**      | **SSIM**  | **FID**  |  **IS**   |
>     | :------: | :---: | :---: | :--: | :---: |
>     | **HR-VITON** |     0.062      | 0.864 | 9.38 | 2.985 |
>     |  **AVTON**   |     0.055      | 0.872 | 8.91 | 3.01  |
>
>     (a) Comparisons for generating try-on images with $256\times192$ resolution
>
>     | **Methods**  |  **LPIPS**      | **SSIM**  | **FID**  |  **IS**   |
>     | :------: | :---: | :---: | :--: | :---: |
>     | **HR-VITON** |      0.061      | 0.878 | 9.90 | 3.093 |
>     |  **AVTON**   |        0.059      | 0.883 | 9.21 | 3.252 |
>
>     (b) Comparisons for generating try-on images with $512\times384$ resolution
>
>     | **Methods**  |  **LPIPS**      | **SSIM**  | **FID**  |  **IS**   |
>     | :------: | :---: | :---: | :--: | :---: |
>     | **HR-VITON** |        0.065      | 0.892 | 10.91 | 3.142 |
>     |  **AVTON**   |    0.076      | 0.887 | 10.31 | 3.154 |
>
>     (c) Comparisons for generating try-on images with $1024\times768$ resolution
>
>     Table B: HR-VITON v.s. AVTON based on different image generation resolutions. we will add the results as new simulations to the revised manuscript.
>
> 4. Regarding the FID criterion to measure the similarity of data distributions between the generated results and the referenced images, we will say we have conducted the simulation results based on VITON and the proposed datasets by utilizing the measurement. The simulation results are as follows. From Table C, we can see that the proposed work can also achieve the best performance than other compared methods in most cases. By taking the reviewer’s comment into account, we will in the revised manuscript to supplement this result.
>
>     | Datasets  | **CP-VTON** | **VTNFP** | **ACGPN** | **PFAFN** | **AVTON** |
>     | :------: |  :-----: | :---: | :---: | :---: | :---: |
>     |  **VITON**    |  24.43  | 17.32 | 15.67 | 6.42  | 5.82  |
>     | **Zalando**   |  26.20  | 18.73 | 16.98 | 6.50  | 5.88  |
>
>     Table C: FID Metrics, we will update the results. We will update the results to Table 1 and 2 in the revised manuscript.
>
> 5. Regarding the grammar and writing errors, we in the revised manuscript have carefully checked the whole manuscript in order to not make any typos or mistakes.
>
> We will share the code and datasets in near future.

---

> ### Author Response · Authors · 2022-11-14
> **Response NW3C_part1**
>
> We must thank for the reviewer’s comment.
>
> 1. Regarding the difficulty of arbitrary try-on task, we will say the arbitrary try-on task can be divided into one-category try on task and cross-category try on task. The one-category try-on task means trying-on within the same category (e.g., long sleeve<-> long sleeve or long pant<->long pant). This has been study by many works and achieve satisfied results; to the contrast, the cross-category try-on task is to try on between different categories (e.g., long sleeves<->short sleeves or long pants<->skirts or coat<-> whole-body dress etc.).
> However, when training the models, we found that is the main difficulty or challenge in the arbitrary virtual try-on system is cross-category try-on. A case in point is that when people aim to try on from long sleeves to short sleeves, some parts of people’s arms will be exposed. Therefore, it is necessary to preserve the characteristics of the reference person and predict such an exposed human body when generating the image-realistic try-on results. However, most current methods mainly focus on try-on with one category by preserving the characteristics of clothes but do not consider the limb prediction when handling try-on with different categories. As a result, some bad try-on performances, e.g., the limbs of human beings are covered by clothes, the color of the skin is wrongly painted and the hand details cannot be properly generated, may be appeared. From this point of views, we have to say arbitrary try-on task can really be realized if the model can well handle cross-category try-on task.
>
>     - To handle above problems, we in this work indeed have developed a special-designed module, namely, Limbs Prediction Module, for predicting limbs, and keep the head and the non-target human body parts, to preserve the characteristics of the reference person. This module is especially suitable for handling cross-category try-on task, such as long sleeves<->short sleeves or long pants<-> skirts, etc., where the exposed arms or legs (including their skin colors and details) can be reasonably predicted. This is good to help the try-on system for formulating a realistic result in the following modules. Ablation study has verified the effectiveness of this module.
>
> 2. Regarding details of the Zalando-Dataset dataset, we will say that the Zalando-Dataset is crawled from https://www.zalando.co.uk/, which is a public and famous website providng clear, extensive and large-scale clothes images for fashion analysis. Here, our collected dataset contains 34928 frontal-view human (man and woman) and clothing (top, bottom, and whole) image pairs. In our study, we split it into training and testing set with 32746 and 2182 image pairs, where the training set contains 19185 tops, 10587 bottoms, and 2974 whole clothes, while the testing set contains 1310 tops, 692 bottoms, and 180 whole clothes.
>
>     - In addition, we also make some comparison work between VITON dataset and the Zalando dataset. Firstly, we make the statistics and distributions of all categories of clothes according to gender and styles, where images of top clothes take up 58.68\% while those of bottom and whole clothes take up remaining 41.32\%. Since the conventional VITON-Dataset (available at https://github.com/minar09/ACGPN) only involves images with top clothes while lacks of those of bottom and whole clothes, we will say the collected Zalando-Dataset is an extension to VITON dataset by involving more clothes categories, which is good for handling real-world arbitrary try-on task;
>
>     - In order to further show the superiority of the collected dataset, we will analyze the characteristics of the dataset and compare with VITON dataset. Here, we choose two measurements for comparisons: 1) the size of dressed clothes taking up the whole referenced image, and 2) the complexity scores for referenced image (only for TOP), which are firstly defined in ACGPN and are divided into easy, medium and hard cases.
>
>         - By analyzing the size, we found that the clothing size in VITON-Dataset is concentrated around 0.4 (which means that the clothing occupies 2/5 of the image). While the clothing size distribution in Zalando-Dataset is between 0.3$\sim$0.5 (that is, the clothing accounts for 3/10$\sim$1/2 of the image). This means that the collected dataset has different sizes of dressed clothes and is more extensive and general in data selection. This can also be more realistic and closer to the real-world applications. As a result, a try-on method that can handling different size of dressed clothes is more useful and practical, which is good to satisfy the requirement of user.
>
>         - By analyzing the complexity, we found that the Zalando-Dataset is higher than that of VITON-Dataset. Therefore, a method trained on such dataset can handle more complex try-on task such as Limbs’ intersections and torso occlusions, which is good to enhance the robustness and adaptiveness of the model.

---

### Decision · Program_Chairs · 2023-01-20

**Decision:**

Accept: poster

**Justification For Why Not Higher Score:**

See the discussions between AC and reviewers. Recent literature and comparisons were missing at submission time.

**Justification For Why Not Lower Score:**

Experiments with latest work at rebuttal time changed the rating of a reviewer, and the committee was a little bit on the side of accepting the work.

**Metareview: Summary, Strengths And Weaknesses:**

This paper studies the virtual try-on of clothes from images. The paper received mixed reviews. The work is well-written. The method proposed a type of part-based model for virtual try-on and achieved good results. The major issue is that the original paper did not discuss/compare with latest progress on the problem. There are extensive discussions between reviewers and the AC. The rebuttal, which was submitted on time, provided comparisons to some recent works, which has convinced a borderline rejection reviewer to increase its score.

**Note From Pc:**

if the above contains the word "oral" or "spotlight" please see: "oral" presentation means -> notable-top-5% and "spotlight" means -> notable-top-25%. As stated in our emails, we are disassociating presentation type from AC recommendations

**Summary Of Ac-Reviewer Meeting:**

The committee members had extensive discussions on this paper. The major issue is whether there are sufficient citations/comparisons with latest works. The reviewers checked the provided comparisons with more recent works at rebuttal time. In the end, the committee was a little bit inclined to accepting the work.